

# Effect of rare earth oxide labeling and sieving methods on aggregate turnover and carbon dynamics

Yike Wang [1]*, Asano Maki [2], Qi Jiang[1], Kenji Tamura [2]

[1]Graduate School of Science and Technology, University of Tsukuba, 1-1-1 Tennodai, Tsukuba, Ibaraki 305-8572, Japan

[2]Faculty of Life and Environmental Sciences, University of Tsukuba, 1-1-1 Tennodai, Tsukuba, Ibaraki 305-8572, Japan

*Correspondence to*: Wang Yike (wyk5569@gmail.com)

**Abstract.** Rare earth element oxides (REOs) are effective tracers to investigate soil aggregate dynamics and are also useful to quantify the interaction between C and aggregate dynamics. Although the effect of the REO labeling process on soil aggregates has been considerably investigated, its effect on soil organic carbon remains unknown. The objectives of this study were to (1)

determine the effect of the labeling process on soil organic matter, (2) verify the feasibility of using REOs as tracers for investigating Andisols dry and wet sieving aggregate turnover, and (3) analyze the relationship between organic matter and aggregate dynamics during 28 days of incubation. The results showed that the soil organic carbon pool was interfered with by the labeling process, particularly with dissolved organic carbon (DOC), microbial biomass carbon (MBC), and free particulate organic matter (fPOM). Furthermore, the degree of interference was related to the soil sieving method, with the wet sieving

process exerting a more significant effect on MBC and fPOM, and the dry sieving process biasing toward DOC. The close 1:1 relationship between measured aggregates and model predictions revealed that REOs are effective tracers for investigating both dry and wet sieving aggregate dynamics in Andisols. Regarding the relationship between organic matter and aggregate dynamics, dry sieving macroaggregate breakdown and restabilization were the largest, shortly appearing in the first incubation week and slowing down thereafter. This trend was also applicable for each dry sieving fraction turnover rate, which correlated significantly

with fPOM (0.97, 0.99, and 0.997, $P < 0.05$). The turnover of wet sieving aggregates also occurred primarily in the first 7 days, but no significant relationship was observed between wet sieving aggregates and soil organic matter dynamics ($P > 0.05$), which was attributed to numerous wet–dry cycles during the labeling process. The results of the current study indicate that dry sieving aggregates fit better with the quantification of the relationship between aggregates and organic matter dynamics when soil organic matter dynamics were quantified using soil organic carbon pools.

## 1    Introduction

Soil structure is a crucial ecosystem service essential for maintaining physical, chemical, and biological processes in the soil. Soil aggregates, clusters of soil particles that adhere to soil organic components, are the fundamental units of soil structure. Soil aggregate dynamics involve aggregate formation, stabilization, and breakdown processes (Oades, 1993; Six et al., 2004). However, evidence shows that soil structure is never in a stable state but is constantly changing (Leij et al., 2002). Studies have

suggested that this dynamic behavior mediates, to a large degree, soil organic matter (SOM) turnover and storage (Six et al., 1998; Plante and McGill, 2002). Although considerable studies have investigated soil aggregate dynamics, the life cycle of an aggregate and its impact on microbial-mediated C cycling remain elusive. Hence, quantifying soil aggregate turnover and SOM dynamics is essential to advance the understanding of SOM dynamics (Plante et al., 2002; De Gryze et al., 2006; Peng et al., 2017) and provide a better prediction of the response of the soil system to management practices.



Studies conducted over decades have demonstrated that understanding SOM aspects by studying and modeling it as a single, uniform entity (Jenkinson, 1990; Parton et al., 1988; Trumbore, 2009) is difficult, and a widespread agreement on the need to separate total SOM into components with contrasting behavior was noted. Before the availability of advanced spectroscopic methods in the early 1990s, SOM studies required the separation of the organic from the mineral phase through chemical extraction procedures. Based on alkaline extractions, "humus" became widely adopted, and "humic substances" were universally

accepted as experimental proxies for SOM. However, these "humic substances" have not been observed using modern analytic techniques. In the place of chemical separations, conceptualizing soil as discrete C pools with differing turnover times has gained favor (Cambardella and Elliott, 1992; Christensen, 2001; von Lützow et al., 2008). The dynamics of soil organic carbon (SOC) and its relationship with soil structure are more often compartmentalized into four soil carbon pools, viz., unprotected, physically protected, chemically protected, and biochemically protected (Six et al., 2002). Based on this, the measured SOC fractions that

represent these pools are determined using both physical and chemical fractionation procedures to provide data to populate the carbon pools in SOC models (Zimmermann et al., 2007).

The interaction between SOC and soil structure has been extensively discussed in the literature (Edwards and Bremner, 1967, Krull et al., 2004, Kleber et al., 2007). Most previous studies have concentrated on establishing how the inherent chemical properties of SOC bind elementary inorganic particles into soil aggregates (Edwards and Bremner, 1967, Amézketa, 1999,

Kleber et al., 2007) and how the resulting organomineral associations can preserve SOC from further degradation (Hassink, 1997). However, SOC stabilization through physical protection by soil aggregates, both macroaggregates and microaggregates, is also significant (Golchin et al., 1994, Jastrow, 1996, Six et al., 2000). Nevertheless, limited studies were noted to directly describe this relationship, in particular, between aggregate formation, stabilization, and breakdown processes.

A useful method for the quantitative analysis of soil aggregate dynamics has been addressed to some extent using tracers. A

series of tracers have been used over the previous 30 years, such as isotopes (Ritchie, 1990; Wallbrink, 1993), Au and Ag (Olmez et al., 1994), and exotic particles (Staricka et al., 1992; Plante et al., 1999; Plante and McGill, 2002). These tracers were replaced by rare earth element oxides (REOs) in subsequent studies due to their small particle size (<5 μm), similar physical and chemical properties, low background concentrations, harmless to the environment, limited mobility and low solubility, and high analysis sensitivity, which enable convenient measurement and low cost, providing excellent properties as a potential mineral tracer

(Tyler, 2004; De Gryze et al., 2006; Peng et al., 2017). In previous studies, Liu et al. (2019) and De Gryze et al. (2006) demonstrated the effects of different labeling methods on different soil types, and subsequently, REOs and carbon isotope dual-labeling methods were used to quantify the relationship between soil aggregates and SOM dynamics (Peng et al., 2017; Wang et al., 2020; Halder et al., 2022). However, REOs have been used in limited soil types, such as Alfisol in the studies of De Gryze et al. (2006) and Morris et al. (2019); Ultisols, Mollisols, and Inceptisols in the studies of Liu et al. (2019); and Vertisols in the

study of Rahman et al. (2019). Andisols, the major soil type in Japan, are characterized by high contents of short-range-order (SRO) minerals and organometal complexes, very low bulk density, and a physically stable aggregate structure (Takahashi et al. 2016). A lack of study still exists on the use of REOs as tracers to label Andisols and the effect of the labeling process on



Andisols organic matter.

Therefore, this study aims to (1) determine the effect of the labeling process on SOM, (2) verify the feasibility of using REOs as tracers for investigating Andisols dry and wet sieving aggregate turnover, and (3) analyze the relationship between organic matter and aggregate dynamics during 28 days of incubation. Using REOs as tracers for labeling Andisols could provide quantitative data to improve the understanding of the dynamics of soil aggregates. Moreover, exploring the effects of labeling and incubation processes on SOC pools provides the possibility to analyze the relationship between soil aggregates and organic matter dynamics. The present study could provide fundamental knowledge for choosing a sieving method based on the coupled processes of Andisols aggregation and SOM dynamics.

## 2 Materials and methods

### 2.1 Soil characteristics

The soil used in this study was sampled from a typical and one of the most well-characterized allophanic Andisols from the experimental field of the National Institute for Agro-Environmental Sciences in Tsukuba (TKB), Honshu Island, Japan (36° 01′ N, 140° 07′ E, 21 m a.s.l). This area has a typical subtropical maritime monsoon climate with annual rainfall and mean temperature of 1,282.9 mm and 13.8°C (during 1981－2010), respectively. The soil is derived from volcanic ash deposits and classified as Silandic Hydric Andosols in WRB (IUSS Working Group WRB, 2006), with the mineralogy being dominated by SRO minerals (allophane/imogolite and ferrihydrite), gibbsite, kaolinite, chlorite, hydroxy-interlayered vermiculite, mica, cristobalite, quartz, and feldspar (Asano and Wagai, 2014). The soil samples were collected from 0-20 cm surface layers by a core at five random locations to prepare one composite sample, which was air-dried and broken by hand to pass through a 2-mm sieve.

### 2.2 REO-labeled aggregates

Three REOs (lanthanum oxide, $La_2O_3$; samarium oxide, $Sm_2O_3$; neodymium oxide, $Nd_2O_3$; and gadolinium oxide, $Gd_2O_3$) were purchased from FUJIFILM Wako Pure Chemical Corporation, Osaka, Japan. The median diameter of the powder (D50) ranged between 1.2 and 3.6 μm. The particle density of the material ranged between 6.5 and 7.6 Mg m$^{-3}$. The background levels of these rare earth elements in the soil were 20.00, 22.00, 4.70, and 4.80 mg kg$^{-1}$ for $La_2O_3$, $Nd_2O_3$, $Sm_2O_3$, and $Gd_2O_3$, respectively.

The REO labeling process has been described elsewhere (Peng et al., 2017). Each REO tracer was added by wet mixing to a separate batch of bulk soil (<2 mm) at a rate of 300 mg kg$^{-1}$, in addition to the control without adding a tracer.The soil was continuously mixed and sprayed slowly with the REE oxides suspended in water. The labeled soil was then stored at 4°C for 7 days to allow water equilibration with minimal microbial activity, followed by incubation at 25°C for 7 days. Next, the soil was oven-dried at 40°C for 48 h and broken down by hand to pass through a 2-mm sieve.

The soil sample measuring <2 mm was separated into three fractions using both wet and dry sieving methods. The wet sieving method was performed as described in a previous study (Elliott, 1986). Briefly, the labeled soil was placed on a filter paper in a mesh, and distilled water was added along with the edge of the filter paper until the soil sample was saturated. The soil sample




was then placed into the top sieve of each set and rapidly immersed in distilled water while being oscillated for 2 min at a displacement of ~3.00 cm at 30 rounds per minute. Water-stable aggregates of 2.00–0.25, 0.25–0.053, and >0.053 mm were obtained. After sieving, the soil aggregates of each size were collected, oven-dried at 40°C, and weighed.

The dry sieving method was performed according to Sainju et al. (2003). Briefly, the labeled soil samples were placed on the sieve, the sieve group was set according to the pore size, and the sieve was shaken by hand for 10 min. Three different dry sieve soil aggregates measuring 2–0.25, 0.25–0.053, and >0.053 mm were separated.

## 2.3 Experimental design

A series of experiments were conducted in this study. First, the effect of the labeling process on SOC pools was determined and the feasibility of using REOs was then verified as tracers for Andisols dry and wet sieving aggregate turnover. Finally, the changes in organic matter and aggregate dynamics were investigated during 28 days of incubation.

### 2.3.1 Effect of the labeling process on SOC pools

Five treatments were designed as follows: (1) soil without REO labeling and sieving processes (background treatment, BG), (2) soil with dry sieving and REO labeling (REO-labeled and dry sieved treatment, REO-D), (3) soil with wet sieving and REO labeling (REO-labeled and wet sieved treatment, REO-W), (4) soil with dry sieving but without REO labeling (dry sieved treatment, CK-D), and (5) soil with wet sieving but without REO labeling (wet sieved treatment, CK-W). These treatments were established by placing 40 g of soil in a 100-ml plastic box, and 13 ml of ultrapure water (UPW) was added to the soil. The effect of REO addition, labeling, sieving, and reconstitution on the organic matter fractions was analyzed by measuring the SOC fractions after 7 days of moisture equilibration at 4°C.

### 2.3.2 Soil incubation

The REO-labeled soil samples were incubated for 28 days at 25°C, with the moisture maintained at 60% water-holding capacity by regularly adding UPW to maintain a constant weight. The aggregate dynamics and SOC fractions were measured by destructively harvesting batches of soil on 0, 7, 14, 21, and 28 days of incubation. On the one hand, the feasibility of using REOs as tracers for aggregate turnover process was verified, and, on the other hand, the relationship between aggregate turnover and organic matter dynamics was analyzed. All treatments were replicated three times.

## 2.4 Measurements

### 2.4.1 Extraction and measurement of REOs

The concentration of REOs in the separated aggregate fractions was measured following Zhang et al. (2001). Exactly 1 g of each aggregate fraction was added to a 50-ml digestion tube. To each tube, 2.5 ml of concentrated HCl (37%) and 7.5 ml of HNO$_3$ (65%) were added and left to react overnight. The samples were then heated at 140°C for 2 h. After cooling to room temperature, the tubes were placed in an ice bath, and 5 ml of H$_2$O$_2$ (10%) was added and left to react overnight. This step was performed to



break up any metal–organic matter complexes. The next day, the samples were heated at 155°C until ~2 ml was left (for ~1 h).

After cooling, the suspension was diluted to 50 ml using UPW and homogenized by shaking. The samples were then diluted thrice due to high concentrations of REOs, followed by filtration through 0.22-mm pore-size membrane filters. The concentrations of rare earth metals in these solutions were measured using a plasma emission spectroscopic analyzer (Shimadzu, ICPS-8100, Kyoto, Japan).

### 2.4.2    Analysis of SOC fractions

The formation of SOM is extremely complex (Haynes, 2005). This complexity makes it difficult to understand SOM dynamics when the soil is subjected to different management systems. Therefore, chemical and physical fractionation methods have been developed to evaluate SOM by examining its fractions. The various fractions were determined using different methods as described in subsequent sections.

#### 2.4.2.1    Total SOC

A 0.15–0.20 g soil sample, which was derived from a 3- to 5-g subsample ground using a ball mill to pass through a 100-mesh sieve, was combusted using a Sumigraph NC-220F Carbon Analyzer set at 870°C, and the content of SOC was evaluated directly by measuring the $CO_2$ content using infrared detection (Wang and Anderson, 1998).

#### 2.4.2.2    Microbial biomass carbon

Microbial biomass carbon (MBC) was determined using the chloroform fumigation–incubation method (Jenkinson and Powlson,
1976). Fumigated and nonfumigated samples were incubated for 24 h at 25°C at a constant moisture content. Microbial C was extracted from both fumigated and nonfumigated samples using 0.5 M $K_2SO_4$. The amount of $CO_2$–C thus evolved was estimated using the method of Snyder and Trofymow (1984). Microbial C content was calculated by subtracting the extracted C content in nonfumigated samples from that measured in fumigated samples (Joergensen, 1996). The values of MBC were represented in micrograms per gram of dry soil.

#### 2.4.2.3    Dissolved organic carbon

Water-extractable organic matter refers to dissolved organic carbon (DOC), which can theoretically be extracted by shaking a soil or sediment in an aqueous medium (modified from Zsolnay, 2003). An aliquot of each 2-mm sieved soil sample was added to a 10-mM $CaCl_2$ solution at a solution–soil ratio of 2:1 and gently stirred for ~10 min at room temperature. The suspension was then centrifuged at 4,000 rpm for 10 min and filtered through a 0.4-μm pore polycarbonate filter. The solutions were then
acidified to pH 2 using 2 M HCl before DOC analyses. This pH value was considered to be the standard condition in the spectroscopic analyses to avoid possible artifacts after further pH adjustments.

#### 2.4.2.4    Particulate organic matter and heavy fraction

The physical fractionation of soils combined both particle size and density fractionation to obtain free particulate organic matter



(fPOM), occluded particulate organic matter (oPOM), and a heavy fraction (HF). Briefly, 4 g of air-dried aggregates were added to a centrifugation tube containing 30 ml of sodium polytungstate solution (SPT; Sometu, Berlin, Germany) with a density of 1.6 g cm$^{-3}$. The tube was inverted gently by hand five times and allowed to settle for 30 min. The solution was then centrifuged at 3,000 rpm for 15 min. The supernatant containing light (<1.6 g cm$^{-3}$) floating particles was filtered (0.45 μm) under vacuum and washed with UPW on a 0.45-μm mesh until the electrical conductivity decreased to <0.50 mS cm$^{-1}$ to obtain fPOM. The residual soil material was dispersed ultrasonically (Branson 250) with an energy of 150 J ml$^{-1}$ for 3.75 min to break down the aggregates (Maki Asano et al., 2018). The fraction of particulate organic carbon released by this energy relative to the total amount was considered as the measure of aggregate stability. The dispersed soil was resieved at 0.45 μm, washed, and dried as described earlier for fPOM. To remove the SPT from the soil, the residue, observed as HF, was rinsed thrice with UPW. The particulate organic matter and HFs were dried to a constant weight at 40°C.

### 2.5 Mathematical description

#### 2.5.1 Calculation of soil aggregate turnover

The mathematical description of aggregate turnover is considered a multidimensional extension of a simple first-order linear compartment model (De Gryze et al., 2006). Briefly, the current study tracked three different aggregate fractions, viz., A (0.25- to 2-mm fraction), B (0.053- to 0.25-mm fraction), and C (<0.053-mm fraction). The three aggregate fractions can produce three breakdown pathways from larger aggregate fractions into smaller aggregate fractions (a–c) and six buildup pathways from smaller into larger aggregate fractions (d–f). The turnover paths and the turnover rate for each aggregate fraction were calculated as shown in **Appendix S1.**

#### 2.5.2 Calculation of SOC pools

The carbon pool rate (in percent) was calculated as follows:

$$\% \text{ of SOC} = \frac{C_i}{TOC}$$

where $C_i$ represents the carbon content of MBC, DOC, fPOM, oPOM, and HF, respectively; TOC is the total organic carbon measured in Sect. 2.4.2.1; and C is the concentration in micrograms per gram of soil.

### 2.6 Statistical analysis

Analysis of variance using the entire data set was performed at each sampling date to determine the effects of REO addition, SOC pools, and their interaction on the measured variables (aggregate size distribution, aggregate turnover pathways, aggregate turnover rate, and SOC fraction content; SPSS, 2004). The least significant difference (at $P < 0.05$) test was applied to evaluate the differences among the mean values of three replicates ($n = 3$). The Pearson correlation test was performed to examine the relationship between the aggregate turnover rate and SOC pools.

### 3 Results



### 3.1 Effect of labeling and sieving processes on SOM fractions

During wet labeling, sieving, and recombining, the SOC pools of bulk soil samples were divided into two different types, disturbed and undisturbed (Table 1). The different treatments had no significant impact on TOC compared with BG ($P > 0.05$), and a similar result was also observed for oPOM and HF, which were considered as slow turnover carbon pools (O'Rourke et al., 2015).

Table 1. The mass content of organic matter fractions in different treatments

*C* concentration in micrograms per gram of soil

| Samples | TOC | MBC | DOC | fPOM | oPOM | HF |
|---|---|---|---|---|---|---|
| BG | 48,880.94 (637.76)[a] | 62.00 (1.50)[a] | 50.93 (1.00)[a] | 1,945.22 (298.64)[a] | 2,543.59 (380.54)[a] | 36,162.91 (211.75)a |
| CK-D | 49,564.36 (265.32)[a] | 52.25 (1.60)[b] | 30.05 (4.09)[b] | 1,034.76 (252.48)[b] | 2,665.78 (505.14)[a] | 36,714.17 (216.10)a |
| CK-W | 49,587.46 (32.43)[a] | 38.61 (1.95 )[c] | 44.75 (1.13)[a] | 371.80 (16.17)[c] | 2,622.14 (88.00)[a] | 35,439.00 (146.98)[a] |
| REO-D | 48,442.64 (9.29)[a] | 53.75 (1.90)[b] | 37.52 (1.83)[b] | 987.78 (44.90)[b] | 2,620.24 (165.32)[a] | 36,258.69 (144.36)[a] |
| REO-W | 50,108.35 (426.21)[a] | 39.82 (0.37)[c] | 41.35 (0.14)[a] | 357.41 (33.25)[c] | 2,573.89 (247.55)[a] | 37,072.33 (180.16)[a] |

Different lowercase letters denote significant differences at $P < 0.05$ between different treatments under the same soil organic carbon fractions ($P < 0.05$).

The active carbon pools appeared to reflect the effects of REO labeling, sieving, and recombing processes on organic matter fractions better than the unaffected carbon pools. The addition of REOs exerted no significant effect on SOM fractions. The

MBC, DOC, and fPOM showed no significant differences between REO-W and CK-W treatments ($P > 0.05$); the same result was also observed in the dry sieving aggregate recombination soil, where the MBC, DOC, and fPOM showed no significant differences between REO-D and CK-D treatments ($P > 0.05$).

The sieving method exerted significant effects on the organic matter fraction. Compared with the untreated soil (BG), dry sieving treatments significantly reduced the DOC and MBC content by ~26.32%‒41.00% and 13.31%–15.73%, respectively ($P < 0.05$),

and also the fPOM content by 46.81%–49.22% ($P < 0.05$). The wet sieving treatments exerted stronger effects than dry sieving treatments on MBC and fPOM, with MBC decreasing by 35.77%–37.72% and fPOM decreasing by 80.89%–81.62%, whereas the DOC content in the wet sieving treatment was not significantly different from that in BG ($P > 0.05$).





### 3.2 Aggregate size distribution

The effects of incubation time on dry sieving aggregate size distribution are shown in Fig. 1. Before labeling and sieving (BG),

the 0.25- to 2-, 0.053- to 0.25-, and <0.053-mm dry sieving aggregates comprised 72.05%, 23.75%, and 4.20% of the total soil

mass, respectively. Before incubation (0 days), a slight decrease from 67.04% to 64.74% on day 7 to 63.37% on day 28 was

observed for the 0.25- to 2-mm aggregates, but the changes in other aggregate size fractions were reversed. As the incubation

days proceeded, a slow increase from 25.37% before incubation (0 days) to 29.95% on day 28 was observed for the 0.053- to

0.25-mm aggregates. The incubation process did not significantly change the proportions for the 0.053- to 0.25- and <0.053-mm

aggregates ($P > 0.05$), whereas labeling and sieving processes significantly increased the proportions for the 0.053- to 0.25- and

<0.053-mm aggregates ($P < 0.05$) and significantly decreased the proportions for the 0.25- to 2.00-mm aggregates ($P < 0.05$).

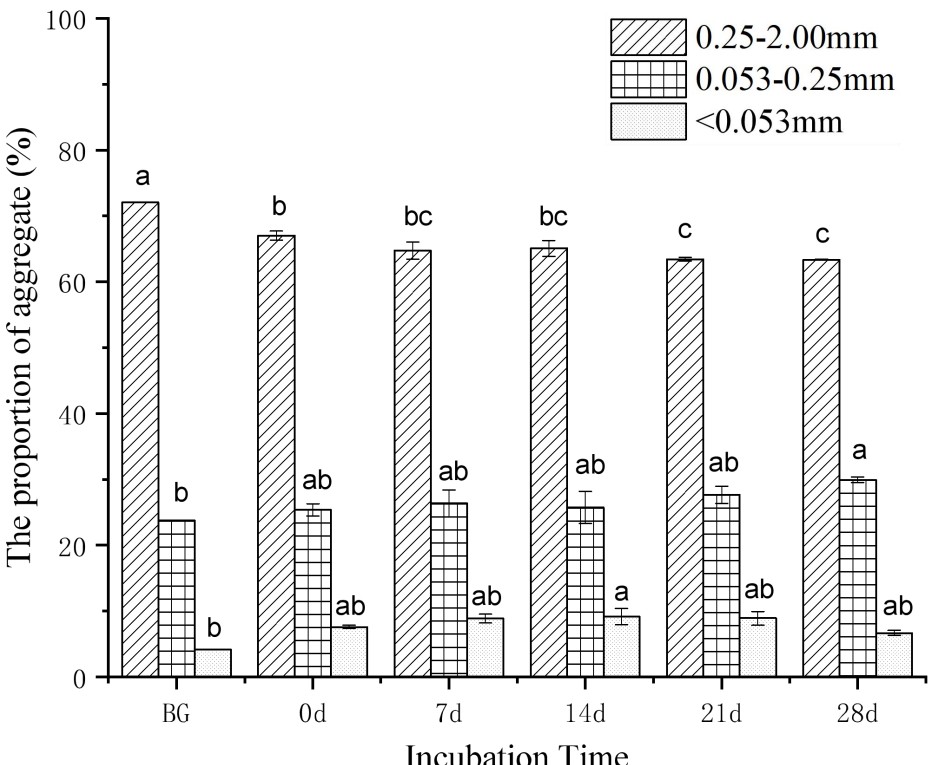

**Figure 1: The proportion of dry sieving aggregate fractions on 0, 7, 14, 21, and 28 days of incubation.** *Different lowercase letters* denote

significant differences at *P < 0.05* between different sampling dates under the same aggregate sizes (*P < 0.05*).

Similar to dry sieving aggregates, no significant effect on the wet sieving aggregates of 0.053–0.25 and <0.053 mm ($P > 0.05$)

was observed during incubation, but a significant decrease from 63.77% before incubation (0 days) to 61.10% on day 7 and a

gradual decrease to 60.35% on day 28 in the 0.25- to 2-mm aggregates was noted. However, labeling and sieving processes

significantly increased the proportions for the 0.053- to 0.25- and <0.053-mm aggregates ($P < 0.05$) and significantly decreased



the proportions for the 0.25- to 2.00-mm aggregates ($P < 0.05$).

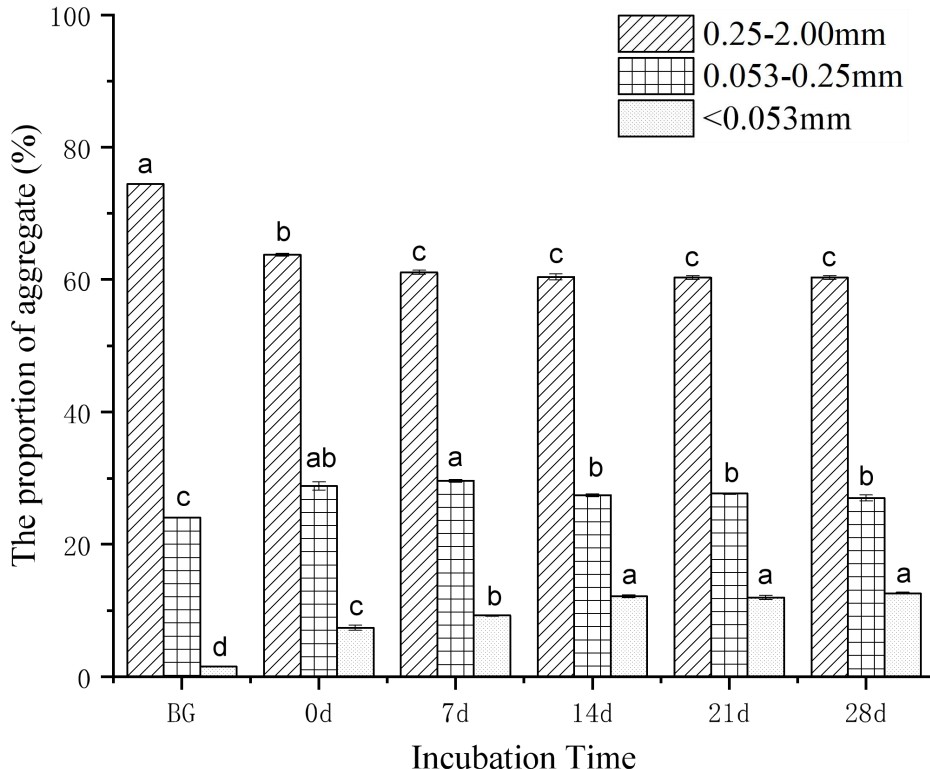


**Figure 2: The proportion of wet sieving aggregate fractions on 0, 7, 14, 21, and 28 days of incubation.** *Different lowercase letters* denote **significant differences at** *P* **< 0.05 between different sampling dates under the same aggregate sizes (** *P* **< 0.05).**

### 3.3. Verification of REOs as tracers for Andisols aggregates

To evaluate the feasibility of using REOs as tracers for investigating the turnover of wet and dry sieving aggregates, the number
of soil aggregates was calculated based on the change of REO contents among the different soil aggregate fractions. These predicted data were compared with the measured soil aggregates during the incubation time (Figs. 3 and 4).



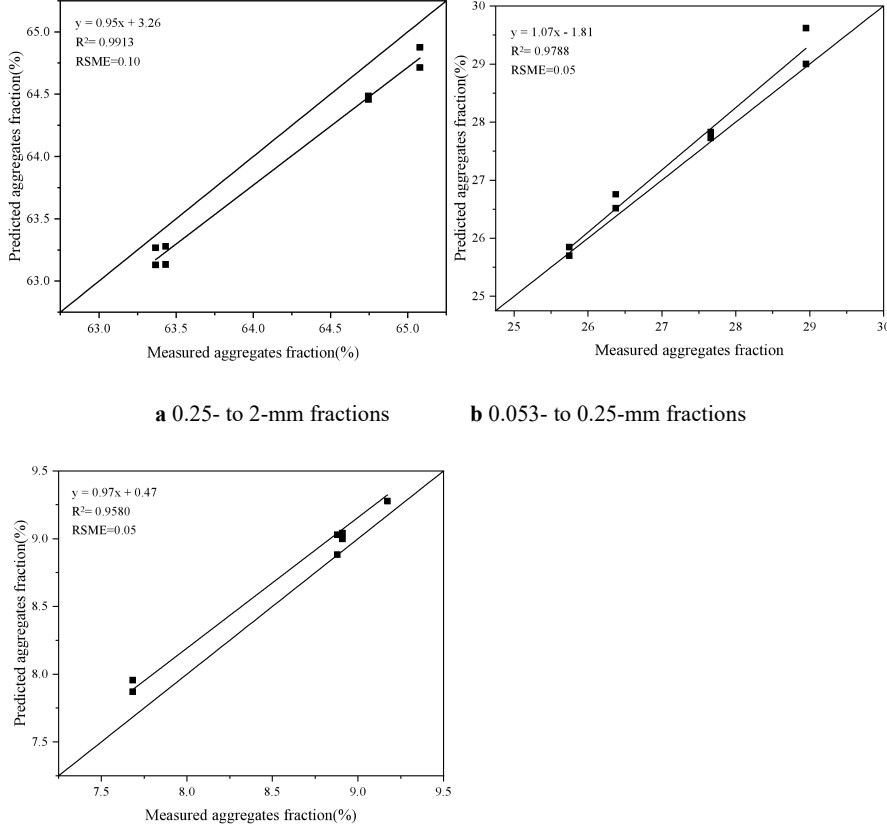

**a** 0.25- to 2-mm fractions          **b** 0.053- to 0.25-mm fractions

**c** <0.053-mm fractions


**Figure 3: The relationship between measured and predicted aggregate percentage using REOs as tracers for investigating the turnover of dry sieving aggregates.**

A significant linear relationship was noted between the predicted and measured aggregate percentages distributed among the dry aggregate fractions. Based on a similar correlation coefficient (*r*) and smaller root mean squared error, good correlations were

observed with the three aggregate fractions and the linear fitted curves. In terms of slope, the close 1:1 line between measured and predicted aggregates supported that REOs were effective tracers of aggregate formation and breakdown dynamics.



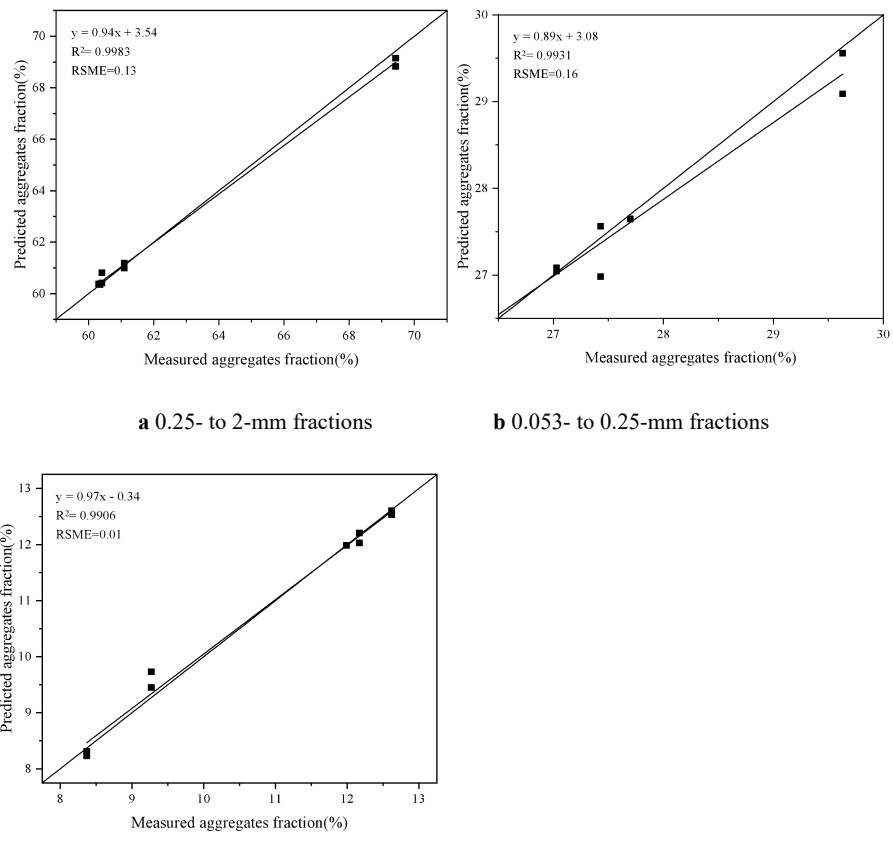

**a** 0.25- to 2-mm fractions          **b** 0.053- to 0.25-mm fractions

**c** <0.053-mm fractions

**Figure 4: The relationship between measured and predicted aggregate percentages using REOs as tracers for investigating the turnover of wet sieving aggregates.**

Compared with dry sieving aggregates, REOs used as tracers were similar in predicting the dynamics of wet aggregates. The close 1:1 line between measured and predicted aggregates occurred among the 0.25- to 2-, 0.053- to 0.25-, and <0.053-mm aggregate fractions. However, for the 0.25- to 2-mm aggregate fractions, the correlation coefficient between the measured values and the fitted curve was only 0.7543, which may be because the variation in the 0.25- to 2-mm fractions was only 60.35%–61.45%, and the REOs have errors in predicting small variations.

### 3.3. Soil aggregate turnover pathways

The transfer pathways among the three aggregate size fractions were determined using the changes in the REO content (Figs. 5 and 6). Before incubation, the recombination and sieving effect caused the aggregates to fragment and recombine. Therefore, the differences in turnover pathways on days 7, 14, 21, and 28 caused by the incubation effects must be calculated after removing the day 0 disturbance.





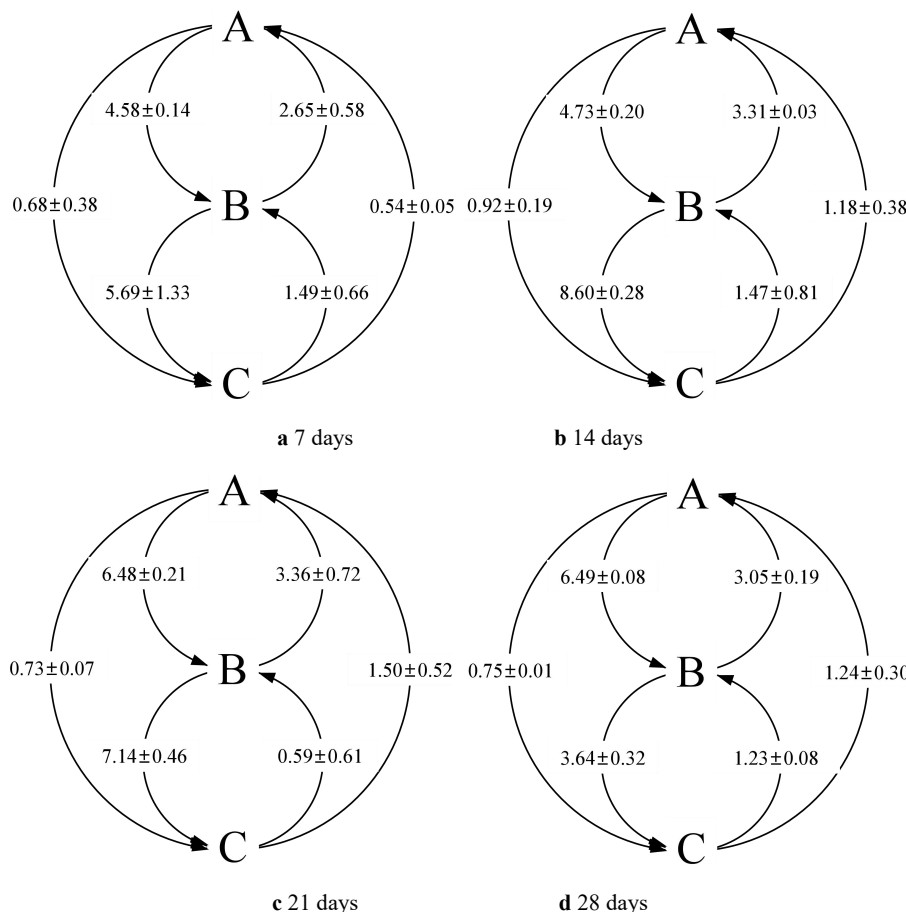

**Figure 5. The six transformation pathways of three dry sieving aggregate fractions on 7, 14, 21, and 28 days of incubation. Values in *arrows* are the relative changes in dry sieving aggregate fractions (in percentage). *A*, *B*, and *C* represent the 0.25- to 2-, 0.053- to 0.25-, and <0.053-mm dry sieving aggregate fractions, respectively.**

Table 2. Turnover rate (in percentages per day) for the three dry sieving aggregate fractions

| Incubation time/day | 0.25 – 2 mm | 0.053–0.25 mm | <0.053 mm |
|---|---|---|---|
| 7 days | $0.75 \pm 0.03$ | $1.19 \pm 0.11$ | $0.29 \pm 0.09$ |
| 14 days | $0.40 \pm 0.01$ | $0.85 \pm 0.02$ | $0.19 \pm 0.08$ |
| 21 days | $0.34 \pm 0.01$ | $0.50 \pm 0.01$ | $0.10 \pm 0.01$ |
| 28 days | $0.26 \pm 0.01$ | $0.24 \pm 0.01$ | $0.09 \pm 0.01$ |

During the incubation period, both breakdown and formation of dry sieving aggregates occurred primarily in the first week and




then aggregate dynamics slowed down in subsequent weeks. Regarding the breakdown process, the 0.25- to 2-mm aggregate

fraction primarily fragmented to form the 0.053- to 0.25-mm fraction, with very little fragmentation to the <0.053-mm fraction,

and this fragmentation was significantly enhanced for 14–21 days, with a significant increase in the 0.053- to 0.25-mm fraction

($P < 0.05$). To evaluate the use of REOs as tracers for polymer transport efficacy, the turnover rate of each polymer fraction (A,

B, and C) was calculated during incubation (Table 2). The turnover rate varied between the different aggregate fractions and

decreased with increasing incubation time. The 0.053- to 0.25-mm fraction had the highest turnover rate of 1.19% per day for the

first 7 days and then decreased to 0.24% per day after 28 days of incubation. The <0.053-mm fraction had the slowest turnover,

with a rate of 0.29% per day in the first week, which decreased to 0.09% per day after 28 days of incubation.

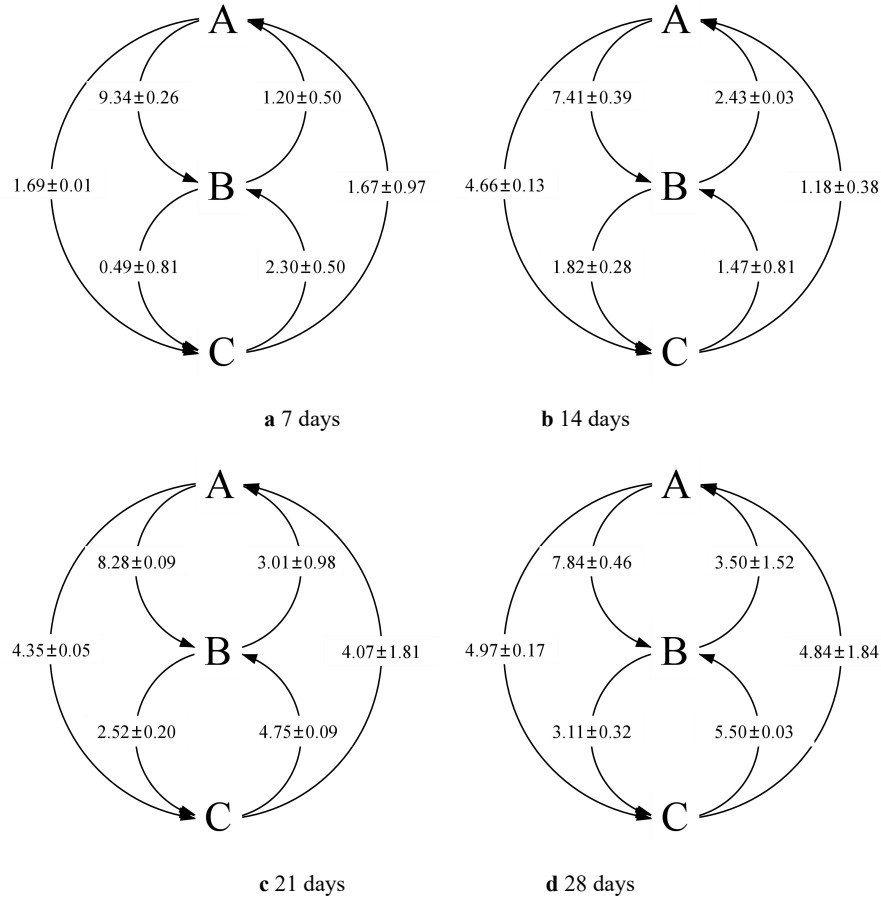

**Figure 6: The six transformation pathways of three wet sieving aggregate fractions on 7, 14, 21, and 28 days of incubation. Values in**
**_arrows_ are the relative changes in wet sieving aggregate fractions (in percentages). _A_, _B_, and _C_ represent the 0.25- to 2-, 0.053- to 0.25-,**
**and <0.053-mm wet sieving aggregate fractions, respectively.**

Table 3. Turnover rate (in percentages per day) for the three wet sieving aggregate fractions



| Incubation time/day | 0.25–2 mm | 0.053–0.25 mm | <0.053 mm |
|---|---|---|---|
| 7 days | 1.58 ± 0.04 | 0.24 ± 0.19 | 0.15 ± 0.21 |
| 14 days | 0.86 ± 0.02 | 0.30 ± 0.03 | 0.32 ± 0.18 |
| 21 days | 0.60 ± 0.01 | 0.26 ± 0.04 | 0.42 ± 0.09 |
| 28 days | 0.46 ± 0.01 | 0.24 ± 0.07 | 0.37 ± 0.06 |

The turnover of the wet sieving aggregates also occurred primarily in the first 7 days, similar to the dry sieving aggregates. However, in contrast to the dry sieving aggregates, the wet sieving aggregates showed more definite sequential patterns of fragmentation. For the breakdown pathways, 9.34% of the 0.25- to 2-mm fractions were broken to constitute the 0.053- to 0.25- mm fractions in the first 7 days, whereas only 1.69% were broken to form the <0.053 mm fractions. However, the turnover from the 0.25- to 2-mm fractions to the 0.053- to 0.25-mm fractions decreased significantly to 7.41%, 8.28%, and 7.84% at 14, 21, and 28 days, respectively, whereas the turnover of aggregates from the 0.25- to 2-mm fractions to the <0.053-mm fractions increased significantly to 4.66%, 4.35%, and 4.97%, respectively. The aggregate turnover rate decreased exponentially over the incubation time (Table 3). During the 28-day incubation, the highest turnover rate was observed for the 0.25- to 2-mm aggregates (1.58%/day), whereas the lowest turnover rate was observed for the <0.053-mm fractions (0.15%/day).

### 3.4. SOC dynamics

Recombined soil was added with UPW to 60% water-holding capacity and equilibrated at 4°C for 7 days as an unincubated sample (0 days). The samples were then incubated at 25°C for 7, 14, 21, and 28 days. The proportions of SOC pools at different incubation days are shown in Fig. 7.

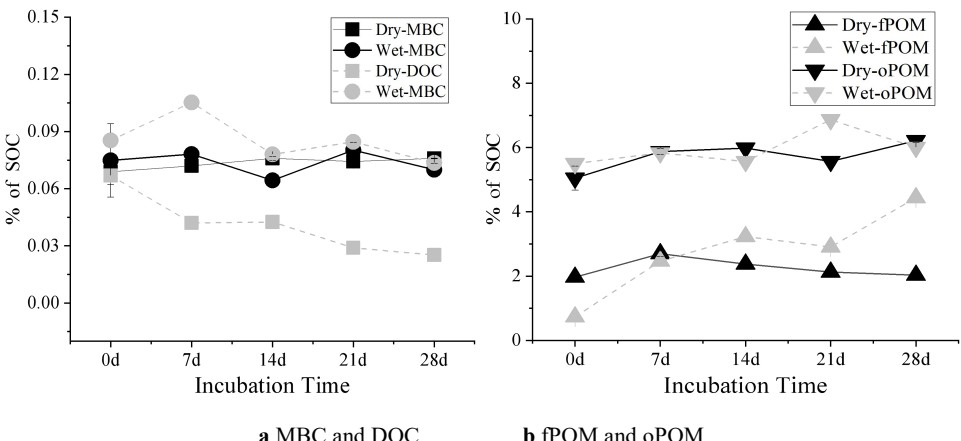

**a** MBC and DOC          **b** fPOM and oPOM



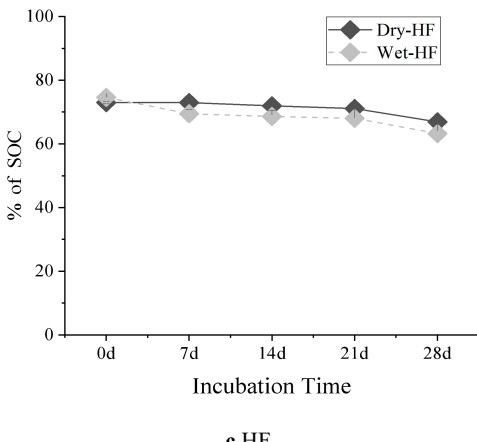

**c** HF

**Figure 7: The proportion of different carbon pools in recombined soils with incubation time.**

DOC and MBC, the small fractions of SOC, remained stable in dry sieving recombined soil at 0.06%–0.08% during incubation. However, in wet sieving aggregate recombined soil, MBC and DOC showed different trends, with DOC first increasing from 0.09% to 0.11% in the first 7 days and then slowly decreasing to 0.07% at 28 days. The MBC decreased to 0.04% and remained in the range of 0.03%–0.04% from 7 to 28 days.

oPOM, which is believed to be protected by aggregates, exhibited an increasing trend throughout the incubation period. The oPOM in dry aggregate recombined soil increased from 5.05% to 6.21% during 28 days incubation, as well as oPOM in wet aggregate recombined soil, increased from 5.50% to 6.00%. The fPOM fraction in dry sieving recombined soil increased from 1.97% to 2.70% through the first 7 days and then slowly decreased to 2.03% from 7 to 28 days. In contrast, for the wet sieving recombined soil, fPOM showed a high growth rate from 0 to 7 days as the fPOM was removed during the wet sieving process, followed by a slow increase to 4.44% on days 7–28. The HF showed remarkable similarity in both recombined soils, declining slowly in the first 21 days and then decreasing sharply after 21 days.

## 4 Discussion

### 4.1 Effects of REO addition and labeling processes on SOM

REOs were commonly used as soil aggregate tracers in combination with 13C to trace the interaction of soil aggregates with SOM (Liu et al.,2019). However, previous studies have not discussed whether the REO labeling process affects SOM. In the present study, the changes in different carbon pools during labeling and recombination were evaluated, which revealed no significant differences in SOC, oPOM, and HF compared with BG. As reported by Six et al. (2002), coarse and fine fPOM, iPOM, and silt- and clay-sized SOM fractions represent unprotected, physically protected, and (bio)-chemically protected SOM pools, respectively. Based on this statement, the unprotected SOM was considered to be mineralized by soil microorganisms and respired into the atmosphere over short timescales, and a portion cycle through the soil slowly, persisting for centuries to millennia before turnover (Dungait et al., 2012). fPOM, DOC, and MBC were classified as SOC pools that respond rapidly to





environmental changes. In this study, the proportions of fPOM, DOC, and MBC were significantly lower in sieved and

recombined samples than in BG samples ($P < 0.05$). REO addition, sieving, and recombining processes may affect a proportion of the SOC pool.

By comparing the different carbon pools with and without the addition of REOs under different sieving methods, the addition of REOs was found to exert no significant effect on the SOC pools (CK-W vs. REO-W and CK-D vs. REO-D, $P > 0.05$). The effect of the addition and type of REOs on aggregates was examined by Liu et al. (2019) by adding REOs to soils of different

properties. However, no evidence of an effect of the REO type and addition on aggregate mass recovery and aggregate distribution was found. Other studies also did not detect an effect of REOs on soil microorganisms (De Gryze et al., 2006; Peng et al., 2017), but their low mobility and solubility were carefully investigated by Zhang et al. (2001). Hence, REOs provide several benefits as tracers for labeling soil aggregates.

Sieving and recombing processes significantly affect the SOC pools. For fPOM, the wet sieving process removes most of the

fPOM and thus reduced the amount of fPOM in soil samples. The dry sieving process also significantly reduced the amount of fPOM due to the removal of fine roots and debris from the soil sample before labeling ($P < 0.05$). The reduction in the MBC and DOC content was related to both moisture content and temperature, where the samples were air-dried during sieving and recombining processes but were stored at 4°C to restrain microbial activity during moisture equilibrium.

### 4.2  REOs as tracers for aggregate turnover

Zhang et al. (2001) determined the distribution of REOs in soils of different particle sizes and similarly found that REO concentrations peaked at 10 – 15 µm fractions, which were ≈ 30% larger than those in bulk soil. To take the REOs tracers uniformly incorporated into different aggregate fractions, Zhang et al. (2001) combined rare earth element powders with soil through serial dilutions and turnover REOs into macroaggregates through 40 days of incubation. However, the prolonged labeling time certainly reduced the SOC content, which may affect the estimation of the relationship between carbon dynamics

and aggregate turnover. Therefore, this study quantified Andisols soil aggregate dynamics by labeling the 0.25-2, 0.053- 0.25, and <0.053-mm aggregate fractions and reduced the incubation time to 7 days to reduce the loss of organic matter during incubation. The close 1:1 line between measured and predicted aggregates (Figs. 6 and 7) demonstrates good labeling effects, which support the viewpoint that REOs are effective tracers of aggregate formation and breakdown dynamics (Zhang et al., 2001; De Gryze et al., 2006; Peng et al., 2017).

### 4.3. Aggregate turnover pathways

Studies have indicated that soil aggregates are in a state of constant change (E Amézketa,1999). The potential importance of aggregate turnover in regulating SOC dynamics has been generally accepted, largely based on the results of evaluating static changes in overall soil aggregation or net changes in aggregate fractions (Yoo and Wander, 2008; Bach and Hofmockel, 2016). The understanding of aggregate turnover pathways, including both formation and breakdown rates, is still limited primarily due

to methodological difficulties (De Gryze et al., 2006; Peng et al., 2017; Morris et al., 2019; Rahman et al., 2019). Using REO



labeling, dry and wet sieving aggregate fractions were found to exhibit different turnover pathways, which are consistent with the trends reported by a few empirical studies (Peng et al., 2017; Wang et al., 2020). The results of this study indicated that the 0.25- to 2-mm dry sieving aggregate fractions tend to break down into microaggregates, which is consistent with Six's hypothesis that microaggregates are created within macroaggregates(Six et al.,2000). Although the wet sieving aggregates were first broken

down into microaggregates in the first week, the formed microaggregates did not appear to be stable, with 1.94% of them still breaking during 7－14 days. The larger microaggregates (20–250 μm) consist of small microaggregates, primary complexes, and occluded OMs whose stability involves persistence and transient binding agents, which result in microaggregates that are still subject to further fragmentation (Oades and Waters, 1991; Oades, 1993).

Simultaneously, according to the REO tracers in the dry sieving aggregates, a portion of the fragmented fraction would turnover

back to the original fraction. For example, 17.48% of the microaggregates broke down to the <0.053-mm fractions after 21 days of incubation, but this proportion decreased significantly to 3.50% when measured at 28 days as reported by Wang et al. (2020). A portion of destabilized aggregates actively participates in aggregate turnover, which may involve organomineral complexes (Christensen, 2002). Individual primary organomineral particles contained in the <0.053-mm fractions and particles of uncomplexed OMs can occur as discrete structural units in the soil, which can incorporate into differently sized aggregates. The

incorporated organomineral complexes encompass mineral-associated and uncomplexed OMs, microorganisms (bacterial colonies, fungal hyphae, and microfauna), and fine roots and thus represent a greater degree of complexity than individual size classes of primary particles (e.g., Oades, 1993; Golchin et al., 1997). However, these dynamics of aggregates with different turnover rates may need to be demonstrated through more detailed quantitative studies.

### 4.4. Relationship between carbon dynamics and aggregation

Soil organic C is closely related to the formation and stability of soil aggregates (Tisdall and Oades, 1982). Aggregates are generally isolated by dry or wet sieving (Kemper and Rosenau, 1986). Dry sieving reflects the stability of aggregates against mechanical disintegration when dry soil is exposed to short periods of rotary sieving in air. In contrast, wet sieving also includes the effects due to the wetting process itself. The mechanical forces imposed on aggregates during wet sieving are less abrasive than those exerted during dry sieving (Emerson and Greenland, 1990). The relationship between soil organic C dynamics and

both dry and wet sieving aggregate turnover is shown in Tables 4 and 5.

Table 4. Correlation matrix between dry sieving aggregates and recombined soil organic carbon pools: A (0.25－2 mm fractions), B (0.053–0.25 mm fractions), and C (<0.053 mm fractions) for aggregate turnover pathways and $v_A$, $v_B$, and $v_C$ for aggregate turnover rate. *MBC* microbial biomass carbon, *DOC* dissolved organic carbon, *fPOM* free particulate organic matter, *oPOM* occluded particulate organic matter, and *HF* heavy fraction for soil organic carbon pools. The numbers displayed within the table

are Pearson coefficients of correlation.

| | A | B | C | $v_A$ | $v_B$ | $v_C$ |
|---|---|---|---|---|---|---|





| | | | | | | |
|---|---|---|---|---|---|---|
| MBC | 0.72 | 0.39 | −0.68 | 0.22 | 0.48 | 0.49 |
| DOC | 0.40 | 0.18 | 0.88 | 0.75 | 0.66 | 0.58 |
| fPOM | 0.95 | 0.15 | 0.35 | 0.97* | 0.99* | 0.997** |
| oPOM | 0.51 | 0.47 | 0.72 | 0.88 | 0.64 | 0.74 |
| HF | −0.97* | 0.24 | 0.27 | 0.93 | 0.994** | 0.99** |

Correlation is significant with **$P = 0.01$ (two-tailed) and *$P = 0.05$ (two-tailed).

In dry sieving recombined soils, as the fPOM content increased, the turnover rate of soil aggregates increased, indicating a significant correlation. This shows that the disintegration of macroaggregates should expose previously protected OM to decomposition. For soils to which the concept of aggregate hierarchy applies, crushing the macroaggregates increases the release of $CO_2$ upon incubation in the laboratory (Christensen, 1996a). The effect of fPOM on soil aggregate turnover rates may be varied. On the one hand, fPOM serves as a nucleus for new aggregates by providing attachment to mineral or organomineral complexes. On the other hand, fPOM provides nutrients to microorganisms and increases their content. Previous studies have shown that microaggregate-associated C plays a vital role in SOC stabilization (Denef et al., 2007). Moreover, Zotarelli et al. (2007) demonstrated that the stabilization of POM in microaggregates is a primary mechanism of SOC protection. In this study, no significant relationship was observed between oPOM and the aggregate fractions of different sizes.

HF is composed of more processed decomposition products that turn over more slowly and have a high specific density because of their close association with soil minerals (Barrios et al., 1996). As the largest SOC pool, HF probably originates primarily from the breakdown of macroaggregates, which in this study correlated significantly with macroaggregate and microaggregate turnover. The majority of soil microorganisms and C and N reside in the HF, which is dominated by soil mineral particles. This can be considered the habitat matrix within which patches of LF/shoot residue and rhizosphere are deposited (Blackwood et al., 2003). A reduction in HF led to a significant increase in the turnover rate of microaggregates and the <0.053-mm fractions.

Table 5. Correlation matrix between wet sieving aggregates and recombined soil organic carbon pools: A (0.25- to 2-mm fractions), B (0.053- to 0.25-mm fractions), and C (<0.053-mm fractions) for aggregate turnover pathways and $v_A$, $v_B$, and $v_C$ for aggregate turnover rate. *MBC* microbial biomass carbon, *DOC* dissolved organic carbon, *fPOM* free particulate organic matter, *oPOM* occluded particulate organic matter, and *HF* heavy fraction for soil organic carbon pools. The numbers displayed within the table are Pearson coefficients of correlation.

| | A | B | C | $v_A$ | $v_B$ | $v_C$ |
|---|---|---|---|---|---|---|
| MBC | 0.84 | 0.85 | 0.997** | −0.80 | −0.51 | −0.28 |
| DOC | 0.89 | 0.90 | 0.62 | −0.92 | 0.31 | −0.94 |



| | | | | | | |
|---|---|---|---|---|---|---|
| fPOM | −0.74 | −0.81 | −0.71 | −0.72 | −0.21 | 0.63 |
| oPOM | −0.48 | −0.41 | −0.61 | 0.42 | 0.26 | −0.12 |
| HF | 0.72 | 0.80 | 0.83 | −0.72 | −0.46 | −0.43 |

Correlation is significant with **$P < 0.01$ (two-tailed) and *$P < 0.05$ (two-tailed).

No significant relationship was observed between the wet sieving aggregate turnover and the organic carbon pool dynamics, which are associated with the sieving and labeling processes of wet sieving aggregates. First, the field bulk soil was air-dried and subjected to wet sieving to remove the >2-mm fractions. The <2-mm fractions were dried, labeled, added with water to 60% water-holding capacity, and incubated for 7 days, followed by second wet sieving, drying, and recombining. Therefore, the labeling process involves two wet sieving and six dry–wet cycles. SOM and aggregate changes caused by dry–wet cycling have been evaluated by Denef et al.(2001). Dry–wet cycling reduces the stability of wet sieving aggregates and prevents the accumulation of fPOM due to the rapid turnover of macroaggregates in soil samples subjected to dry–wet cycling. fPOM correlated significantly with HF. The HF generated by aggregate breakdown cannot accumulate into aggregates, allowing it to accumulate on the outside and consequently increase the proportion of fPOM.

## 5 Conclusion

REOs were demonstrated as effective tracers to explore soil aggregate dynamics and as an effective approach to quantify the interaction between C and aggregate dynamics. The labeling process of REOs interferes with the soil carbon pool, particularly DOC, MBC, and fPOM. The degree of disturbance was related to the soil sieving method, with the wet sieving process exerting a more significant impact on MBC and fPOM, and the dry sieving process biasing toward DOC. The 1:1 relationship between measured and predicted aggregates indicates the effective tracing impact of REOs on both dry and wet sieving aggregates. For dry sieving aggregates, macroaggregate breakdown and restabilization were the largest shortly appeared in the first incubation week and slowed down thereafter. This trend was also verified by the dry sieving aggregate turnover rate. The different fractions of dry sieving aggregates showed the highest turnover rate in the first week, followed by a slow decrease. The turnover rate correlated significantly with fPOM. For wet sieving aggregates, the turnover also occurred primarily in the first 7 days. However, unlike dry sieving aggregates, the microaggregates formed by wet sieving macroaggregate fragmentation broke down again in 7 – 28 days, resulting in the highest turnover rate for wet sieving microaggregates. No significant relationship was observed between wet sieving aggregates and SOM dynamics, which was attributed to numerous wet–dry cycles during the labeling process. Overall, the combined approach of SOC pools and REO labeling of aggregates provides considerable opportunity for further studies to explore the interaction between C and aggregate dynamics in Andisols. In developing this approach, it was found that REO labeling and SOC pools could only be used to trace the general aggregate turnover pathways and organic matter dynamics, which makes it difficult to provide more detailed information. For instance, for a rapid turnover of aggregate fractions, determining the rotation times and the turnover times of the aggregates is difficult.

## 6 Acknowledgements



This work was supported by JSPS KAKENHI Grant Number 21H02086. We would like to thank the China Scholarship Council for support this work through the award of a fellowship to Dr. Wang YK (grant no.202008610192). We also thank Prof.Wagai Rota for providing Andisols to this experiment.

## 7   Competing interests

The contact author has declared that neither they nor their co-authors have any competing interests.

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
