# Peer review of "Effect of rare earth oxide labeling and sieving methods on aggregate turnover and carbon dynamics"

_EGUsphere, 2022_

## Referee Comment (RC2)

Comments to the Author

This manuscript by Wang et al. presented an important and interesting study of the effects of REOs labeling process and sieving methods on aggregate turnover and carbon dynamic. Researching soil aggregates and associated biogeochemical processes is a very time-consuming and laborious work, with the complex REOs labeling process makes it to be more difficult. Recently, studies focusing on aggregate turnover show a rising trend, while less researches have reported the role of labelling and sieving processes played in affecting aggregate turnover and soil C dynamics, which may obscure or even magnify the effect of treatments. Dividing the labeling process from the incubation (or other treatments) thus is pivotal for accurately assess the real aggregate turnover dynamics and soil C dynamics. Indeed, the authors found that labeling process and sieving method affect aggregate turnover and soil C fractions (particularly labile C fraction, i.e., DOC, MBC) more intensively than incubation. However, the text has low readability. The logic is very confusing, and the authors seem unable to catch the highlights and key points of the story. Additionally, some key information about the calculations do not show in article. I very appreciate with the work and its significance the authors done. However, I feel a pity that the authors show the story in a not good way. So, I do not recommend the publication of the article in SOIL. The best result is resubmission after major revision.

Introduction
The introduction does not align well with the topic. The article aims to reveal the effects of labeling process and sieving method on aggregate turnover and soil C dynamics. However, in the introduction, relating statements are rare, and the authors paid more attention to some unrelated points. For example, the soil types previous REOs studies has been used. Is the soil type (Andisols) very important? I do not think so. In my opinion, the authors should show us the shortage in REOs labeling studies (i.e., overlooks the effects of labeling process and sieving method) and its importance in assessing aggregate turnover dynamic and soil C dynamics, possible effects of labeling process and sieving method on aggregate turnover and soil C dynamics and how do them, and the potential relationships between aggregate turnover and soil C dynamics.

L39-42 why do you mention the concept of "humus"? does your study involve the chemical stability of SOM? the POM and HF you studied are fractioned by physical and density fraction method, not by acid or alkaline or thermal hydrolysis, right? Although they have different functions, they are commonly regarded from a perspective of physical stability/protection.

Materials and methods
L81-84: more details of the soil properties show be shown, such as SOC, soil texture and etcs.
L84-85: "soil sampled by a core at five random locations". What's the diameter of soil core? How large the region of soil sampling? Given that soil is highly heterogenous, how representative are the five cores?
L87: you separated three aggregate size classes (0.25-2, 0.053-0.25, <0.053mm), why there have four REOs? One is redundant?
L95: I doubt that you can broke down to pass the soil through a 2-mm sieve just by hand without other tools after oven-dried. The wet-dry cycle of labeling process had clumped the soil.

L114: how do you add the 13 ml of ultrapure water to avoid the rewet effect on aggregate turnover?

L119-120: more details of the incubation. The top of box is open or close?

L175: where is the calculation of aggregate turnover rate? I can not find in the Appendix S1.

Results:

The authors showed lots of information in the text, without emphasizing the important information associated with the topic. For example, the effectivity of REOs labeling in tracing aggregate turnover has been widely proved, it is not an important information here. So, Fig.3 can be put in appendix rather than in text, and associated text should be more concise. Figs. 1 and 2 also can be merged.

L194: why the format of Table 1 differs from other tables? And I feel uncomfortable with the unit in Table 1. g kg$^{-1}$ soil for TOC, fPOM, oPOM and HF and mg kg$^{-1}$ soil for MBC and DOC are more commonly used.

Fig.7: Why do not show the absolute value of soil C fractions such as MBC and DOC? I think use the absolute value is more clearly than the proportions of them in SOC (the values are too low) to assess the effects of REOs labeling and sieving methods.

Discussion:

I completely agree with the comments of another review about the discussion.

Besides, for section 4.3, authors paid more attention to discuss the aggregate turnover dynamics, where is the discussion of comparing the effects of labeling process and sieving method on aggregate turnover? I think it is the key point needing to be discussed.

L334-338: explaining why wet sieving removes most fPOM. Does the dry sieving reduce fPOM because of the removal of fine roots and debris? The fine roots and debris should be removed before experiment, them were not pick out clean?

Some problems of format at L345, 351, 361.

L394-395: If the relationship between oPOM and aggregate fractions is important, please discuss it more in-depth, not just depict the result. If it is not important in this study, it is not necessary to show.

L415: "The HF generated by aggregate breakdown cannot accumulate into aggregates, allowing it to accumulate on the outside and consequently increase the proportion of fPOM", please attach the reference. I do not think the accumulation of HF can increase fPOM. They are varying in size, density and properties (e.g. C/N), two different concepts. The soil continuum model (Lehmann, J., Kleber, M., 2015. The contentious nature of soil organic matter. Nature 528: 60-68) suggested that

fPOM can be degraded into HF with microbial processing, and it is an irreversible process.

Conclusion

Do not simply repeat the results, but show the main findings and implications.

---

## Author Comment (AC1)

**Response to reviewers**

We gratefully thank the editor and reviewer for the time spent making their constructive remarks and helpful suggestion, which has significantly raised the quality of the manuscript and has enabled us to improve the manuscript. Each suggested revision and comment, brought forward by the reviewer was accurately incorporated and considered. Below the reviewer's comments are response point by point and revisions are indicated.

**'Comment on egusphere-2022-728', Anonymous Referee #1, 21 Oct 2022**

**General Comments:** This manuscript from Wang et al. aims at elucidating the effect of rare earth oxide labeling and sieving methods on aggregate turnover and carbon dynamics. To reach their objectives, they conducted labelling and incubation experimentations with dry or wet sieving methods. SOC fractions (DOC, fPOC, MBC and HF) were detected and their relationship with aggregate dynamics were analyzed herein.

Indeed, little researches have reported the effects of labelling and sieving processes on SOC fractions, it is interesting to investigate the discrepancy caused by various methods. But in my view, authors do not provide a clear response to the topic, the research questions are not well stated in the introduction and the findings are not fully discussed in the discussion part. Besides, I doubt the calculation on aggregate turnover rate, which is different from the calculation proposed by De Gryze et al. and Peng et al. For these reasons, I do not recommend the publication of the article in SOIL.

**Response:** We greatly appreciate the reviewer's insightful comments. In fact, while processing the data for this manuscript, we found that there was perhaps a hidden innovation, the application of soil organic fractions to quantify soil organic carbon, with REE oxides to track the aggregate turnover. In previous studies, Peng et al. (2017) analyzed the organic carbon dynamics by adding 13C-labeled glucose to REE oxides labeled soils and determining the 13C content in different aggregate fractions. Subsequently, M. Halder et al. (2022) used eleven organic materials characterized in terms of nutrient stoichiometry, biochemical features and carbon (C) functional groups, to determine which characteristics of organic materials control soil aggregate turnover. However, in the following studies, we found that it would be too expensive to use carbon isotope methods in field experiments or to determine the contribution of organic carbon monomers (e.g. galactosamine) for aggregate turnover. This is why a large number of descriptions of the relationship between aggregates and organic carbon have appeared in previous manuscripts.

The main reason for your query about the calculation is that the transformation paths of three aggregate fractions were divided into (1) turnover directly caused by the labeling and sieving processes (at 0 days incubation); (2) turnover caused by soil

microorganisms during the incubation process (at 7,14,21,28 days incubation). The excessively low transformation of aggregate turnover pathways is due to the subtraction of transformation before incubation (0 days). We will provide a detailed response to your question about the aggregate turnover calculation in a point-to-point response.

Based on your suggestions, we have restructured the logical framework of the manuscript and will respond to your suggestions in a point-to-point response. In the revised version of the manuscript, we have refined the abstract and main text (especially the Introduction, the Results and the Discussion sections) to make the paper easier to read, the procedure for the calculation of aggregate turnover, which was originally placed in the appendix, has also been collated into 2.5.1 Calculation of soil aggregate turnover in the revised manuscript.

In the Introduction section, we have (1) restructured the framework of the manuscript to make the manuscript more palatable to general readers; (2) outlined the major assumptions briefly; (3) deleted unnecessary description of the relationship between aggregate turnover and soil organic carbon dynamics to make the introduction section more relevant to the topic.

In the Material and method section, we have (1) introduced a more specific description of the Andisols soil samples in 2.1 Soil characteristics; (2) Changed the description of the experiment design in 2.3 Experiment design to make it more consistent with the research topic. (3) Added a flow chart of the recombination process in 2.3.1 Recombination process, to make the recombination process more accessible to the readers; (4) Added 2.5.1Calculation of soil aggregate turnover in the revised manuscript, from the original appendix and added Figure 2 The 6 possible transformation pathways of aggregate.

In the Results section, we have (1) modified the structure of the result section according to the revised experiment design, and described the effect of the labeling and sieving process on aggregate turnover and organic carbon fractions, respectively; (2)added transformation aggregates turnover pathways before incubation (0 days) in 3.1.2 Soil aggregate turnover pathways; (3) Added soil organic carbon fraction dynamics of BG treatment during incubation in 3.2.2 The effect of labeling and sieving processes on SOC fractions during incubation process; (4) placed the relationship between aggregate turnover and organic carbon dynamics in 3.3 The effects on the quantitative study of the relationship between aggregate turnover and organic carbon dynamics.

In the Discussion section, we have reorganized the discussion according to the research topic and your comments. the impact of the labeling and sieving processes on soil aggregate turnover and soil organic carbon fractions were discussed, respectively.

In the Conclusion section, We have (1) identified that labeling and sieving processes could affect aggregate turnover and soil organic carbon fractions; (2) made

suggestions for eliminating the disturbances.

**Point to point response**

**Comment 1: Introduction,** The title focused on two factors, namely labelling processes and sieving methods, to aggregate turnover and SOC. Insufficient statements on the importance of these two factors are provided, instead, authors illustrated more the interaction between SOC and soil structure.

**Reply1:** We gratefully appreciate for your valuable comment. In the Introduction section, we have (1) deleted unnecessary description of the relationship between aggregate turnover and soil organic carbon dynamics; (2)Restructured introduction section from labeling and sieving processes on aggregate turnover and organic carbon dynamics, to make the introduction section more relevant to the research topic (L25-L88).

**L25-L88:** Soil structure is a crucial ecosystem service essential for maintaining physical, chemical, and biological processes in the soil. Soil aggregates, clusters of soil particles that adhere to soil organic components, are the fundamental units of soil structure. Soil aggregate dynamics involve aggregate formation, stabilization, and breakdown processes (Oades, 1991; Six et al., 2004). However, evidence shows that soil structure is never in a stable state but is constantly changing (Leij et al., 2002). Studies have suggested that this dynamic behavior mediates, to a large degree, soil organic matter (SOM) turnover and storage (Six et al., 1998; Plante and McGill, 2002). Although considerable studies have investigated soil aggregate dynamics, the life cycle of an aggregate and its impact on microbial-mediated C cycling remain elusive. Hence, quantifying soil aggregate turnover and SOM dynamics is essential to advance the understanding of SOM dynamics (Plante et al., 2002; De Gryze et al., 2006; Peng et al., 2017) and provide a better prediction of the response of the soil system to management practices.

Over the past 40 years, few studies have monitored soil aggregate dynamics using tracers. Indeed, stable isotopes (e.g., 137Cs, 210Pb, 7Be, and 234Th) were first used as tracers to monitor soil aggregate dynamics (Brown, 1981; Ritchie, 1990; Wallbrink, 1993). Then, Olmez (1994) proposed a soil particle labeling method for the diffusion of Au and Ag into sediment crystal lattice by high-temperature heating. However, heating may greatly change the chemical composition of the soil, especially soil clay and soil organic matter, making this method disadvantageous. Others like Staricka et al. (1992) mixed 1-3 mm ceramic spheres into the soil as tracers and found that they incorporated much more rapidly into macroaggregates (>40 mm) than into microaggregates (12-40 mm). However, All these methods have different drawbacks in labeling soil aggregates. Therefore, rare earth element oxides (REE oxides) were proposed to monitor the dynamics of soil aggregates, as they are harmless to the environment and characterized by small particle size (<5  $\mu$ m), similar

physicochemical properties, low background concentrations, limited mobility, and low solubility, resulting in high sensitivity analysis, convenient monitoring results, and low-cost measurement and providing excellent properties as a potential mineral tracer (Tyler, 2004; De Gryze et al., 2006; Peng et al., 2017).

As mentioned above, the REE oxides strongly bind with silty and, in particular, clay soil particles are incorporated into soil aggregates (Zhang et al., 2001), explaining the widespread use of REE oxides as tracers of soil aggregates. Indeed, To obtain reliable results, soil aggregate tracers must be as unaffected and exhibit uniform states in soil. the mean weight diameter (MWD) to determine the De Gryze et al. (2006) used effect of REE oxides addition and labeling processes on soil aggregate, showed no evidence that the tracer affected microbial activity or initial aggregate size distribution. At the same time, calculated the rare-earth element (REE) enrichment percentages in each aggregate size, found the coefficients of variation of the rare-earth element measurements in eight spatially separate samples were acceptably small and the recovery rates large. From this, Liu et al. (2019) showed no evidence of potential effects of REE oxide types and addition on aggregate recovery rate and aggregate distribution in different soil types. Meanwhile, Morris et al.(2019) provided qualitative snapshots of eight REE oxides labeled aggregate thin sections through the X-ray fluorescence data, which also consistent with the results reported in the previous studies. The above studies demonstrate that REE oxides addition and labeling processes have little effect on soil aggregates, but few studies have concerned the effect of REE oxides addition and labeling processes on soil organic carbon dynamics.

Soil total organic carbon (TOC), occluded organic carbon(oPOC) and heavy fraction(HF) are regarded as slow fractions of soil organic carbon because they change slowly over time because of their large contents, protected by aggregates and associated with minerals (Franzluebbers et al., 1999;Riggs et al.,2015;Marín-Spiotta et al., 2008). In contrast, dissolved organic carbon (DOC) and microbial biomass carbon are considered as active organic carbon fractions that change seasonally (Wilson and Xenopoulos, 2008; Babur and Dindaroglu, 2020) and influence aggregate turnover(Murugan et al., 2019; Bucka et al., 2019). Similarly, free particulate organic carbon (POC) are considered as intermediate organic carbon fractions for changes of soil organic carbon with time that provide substrates for microbes (Witzgall et al., 2021) and also influence aggregation (Bucka et al., 2019). Because the aggregate distribution and aggregate turnover influence soil organic carbon (Six et al., 1998), determination of soil organic fractions provides important information on soil organic carbon sequestration and mineralization with aggregate turnover(Qiu et al., 2023).

The soil aggregate and associated carbon (C) fractions, however, are also influenced by the sieving method (Kemper and Rosenau, 1986; Whalen and Chang, 2002). The most common method of separating aggregates from bulk soil is the wet sieving method, and the wet sieving method is widely used to determine size distribution and stability of aggregates caused by raindrop impact on dry soil causing slaking and the gas pressure inside aggregates (Elliott, 1986; Cambardella and Elliott, 1993). Compared with wet sieving method, The dry sieving method is based on the mechanical impact on soil structure during shaking the samples in a sieve tower, either by hand (Bach & Hofmockel, 2014) or using a mechanical sieve shaker(Nahidan & Nourbakhsh, 2018). As a result, dry sieving method for determining microbial biomass and activities that include water-soluble C and N in aggregates is receiving increased attention because of less destruction to the physical habitat of microbial communities(Kooch et al., 2022). Although information on soil organic carbon fractions is available (Kemper and Rosenau,1986), little is known about variations in soil organic carbon fractions with aggregate size fractions separated by dry and wet sieving methods.

On the other hand, REE oxides have been used in limited soil types, such as Alfisol in the studies of De Gryze et al. (2006) and Morris et al. (2019); Ultisols, Mollisols, and Inceptisols in the studies of Liu et al. (2019); and Vertisols in the study of Rahman et al. (2019). Andisols, the major soil type in Japan, are characterized by high contents of short-range-order (SRO) minerals and organometal complexes, very low bulk density, and a physically stable aggregate structure (Takahashi et al. 2016). A lack of study still exists on the use of REE oxides as tracers to label Andisols and the effect of the labeling process on Andisols organic carbon dynamics. Therefore, according to Peng et al. (2019), the study was divided into the labeling - sieving stage and the Recombination - incubation stage, for both short- and long-term effects, aims to the effect of the labeling and sieving processes on soil aggregate assess (1) turnover, (2) the effect of the labeling and sieving processes on soil organic carbon fractions, and (3) the impact on the quantitative analysis of the relationship between organic carbon and aggregate dynamics. Unlike previous studies that have focused on the effects of REE oxides labeling and sieving methods on soil aggregate size fractions, this paper provides the perspective of soil organic carbon fraction for the feasibility of using REE oxides as Andisols aggregate tracers.

**Comment 2:** L80 More details on the investigated soil should be provided, such as the initial SOC content, sand/clay/silt content, bulk density etc.

**Reply2:** We gratefully appreciate for your valuable comment. We provide the soil properties in 2.1Soil characteristics in revised manuscript (*2.1 Soil characteristics*, L98-L101).

**L98-L101:** Prior to the labeling and incubation experiment, the soil properties were: bulk density: $0.78g \text{ cm}^{-3}$ ; soil pH: 6.50; total organic carbon (TOC): 53.21 mg g-1, total N: 4.54 mg g-1, short range order mineral(SRO: allophane + ferrihydrite): 168 mg g-1; Sand (2.0–0.05 mm): 23.92%; Silt (0.05–0.002 mm): 31.02%; Clay (<0.002 mm):

**45.06%.**

Comment 3: L90-100 Four oxides were used for labelling, but only 3 aggregate fractions were used? So which three oxides you used herein? How to get the recombined soil columns? The soil content and bulk density of these recombined columns? How many soil columns in total? More detailed information is needed.

L120 What do you mean by "regularly"? Every two days?

**Reply3:** We feel sorry for the inconvenience brought to the reviewer.

(1) We have placed the labeling process in the 2.3 Experiment design section, together with sieving process, recombination process and incubation process. And the labeling process is described in 2.3.1 Labeling process (L127-L131).

L127-L131: The REE oxides labeling process has been described elsewhere (Peng et al., 2017). Briefly, each REE oxides was suspended in ultra pure water by vortex mixing at a concentration of 300 mg kg-1. 50g soil was continuously mixed and sprayed slowly with the REE oxides suspensions to homogenize labeling, the labeled soil was then stored at 4°C for 7 days to allow water equilibration with minimal microbial activity, followed by incubation at 25°C for 7 days. Next, the soil was oven-dried at 40°C for 48 h and broken down by hand to pass through a 2-mm sieve.

(2) For details of the recombination process is described as 2.3.3 The recombination process and the protocol for recombining aggregates into recombined soils was added, as shown in *Figure 1* (L143-L153).

L143-L153: The recombination process includes the recombination of REE oxides labeled aggregates and the recombination of blank aggregates. For the REO-D and REO-W treatments, the recombined soil consisted of the 2-0.25 mm fraction labeled by Gd2O3 (A), 0.25-0.053 mm fraction labeled by Sm2O3 (B), and

---

## Author Comment (AC2)

**Response to reviewers**

We gratefully thank the editor and reviewer for the time spent making their constructive remarks and helpful suggestion, which has significantly raised the quality of the manuscript and has enabled us to improve the manuscript. Each suggested revision and comment, brought forward by the reviewer was accurately incorporated and considered. Below the reviewer's comments are response point by point and revisions are indicated.

**'Comment on egusphere-2022-728', Anonymous Referee #1, 21 Oct 2022**

**General Comments:** This manuscript from Wang et al. aims at elucidating the effect of rare earth oxide labeling and sieving methods on aggregate turnover and carbon dynamics. To reach their objectives, they conducted labelling and incubation experimentations with dry or wet sieving methods. SOC fractions (DOC, fPOC, MBC and HF) were detected and their relationship with aggregate dynamics were analyzed herein.

Indeed, little researches have reported the effects of labelling and sieving processes on SOC fractions, it is interesting to investigate the discrepancy caused by various methods. But in my view, authors do not provide a clear response to the topic, the research questions are not well stated in the introduction and the findings are not fully discussed in the discussion part. Besides, I doubt the calculation on aggregate turnover rate, which is different from the calculation proposed by De Gryze et al. and Peng et al. For these reasons, I do not recommend the publication of the article in SOIL.

**Response:** We greatly appreciate the reviewer's insightful comments. In fact, while processing the data for this manuscript, we found that there was perhaps a hidden innovation, the application of soil organic fractions to quantify soil organic carbon, with REE oxides to track the aggregate turnover. In previous studies, Peng et al. (2017) analyzed the organic carbon dynamics by adding 13C-labeled glucose to REE oxides labeled soils and determining the 13C content in different aggregate fractions. Subsequently, M. Halder et al. (2022) used eleven organic materials characterized in terms of nutrient stoichiometry, biochemical features and carbon (C) functional groups, to determine which characteristics of organic materials control soil aggregate turnover. However, in the following studies, we found that it would be too expensive to use carbon isotope methods in field experiments or to determine the contribution of organic carbon monomers (e.g. galactosamine) for aggregate turnover. This is why a large number of descriptions of the relationship between aggregates and organic carbon have appeared in previous manuscripts.

The main reason for your query about the calculation is that the transformation paths of three aggregate fractions were divided into (1) turnover directly caused by the labeling and sieving processes (at 0 days incubation); (2) turnover caused by soil

microorganisms during the incubation process (at 7,14,21,28 days incubation). The excessively low transformation of aggregate turnover pathways is due to the subtraction of transformation before incubation (0 days). We will provide a detailed response to your question about the aggregate turnover calculation in a point-to-point response.

Based on your suggestions, we have restructured the logical framework of the manuscript and will respond to your suggestions in a point-to-point response. In the revised version of the manuscript, we have refined the abstract and main text (especially the Introduction, the Results and the Discussion sections) to make the paper easier to read, the procedure for the calculation of aggregate turnover, which was originally placed in the appendix, has also been collated into 2.5.1 Calculation of soil aggregate turnover in the revised manuscript.

**In the Introduction section**, we have (1) restructured the framework of the manuscript to make the manuscript more palatable to general readers; (2) outlined the major assumptions briefly; (3) deleted unnecessary description of the relationship between aggregate turnover and soil organic carbon dynamics to make the introduction section more relevant to the topic.

**In the Material and method section,** we have (1) introduced a more specific description of the Andisols soil samples in *2.1 Soil characteristics*; (2)Changed the description of the experiment design in *2.3 Experiment design* to make it more consistent with the research topic. (3) Added a flow chart of the recombination process in *2.3.1 Recombination process*, to make the recombination process more accessible to the readers; (4) Added *2.5.1Calculation of soil aggregate turnover* in the revised manuscript, from the original appendix and added Figure2 The 6 possible transformation pathways of aggregate.

**In the Results section,** we have (1) modified the structure of the result section according to the revised experiment design, and described the effect of the labeling and sieving process on aggregate turnover and organic carbon fractions, respectively; (2)added transformation aggregates turnover pathways before incubation (0 days) in *3.1.2 Soil aggregate turnover pathways*; (3) Added soil organic carbon fraction dynamics of BG treatment during incubation in 3.2.2 The effect of labeling and sieving processes on SOC fractions during incubation process; (4) placed the relationship between aggregate turnover and organic carbon dynamics in *3.3 The effects on the quantitative study of the relationship between aggregate turnover and organic carbon dynamics*.

**In the Discussion section,** we have reorganized the discussion according to the research topic and your comments. the impact of the labeling and sieving processes on soil aggregate turnover and soil organic carbon fractions were discussed, respectively.

**In the Conclusion section,** We have (1) identified that labeling and sieving processes could affect aggregate turnover and soil organic carbon fractions; (2) made

suggestions for eliminating the disturbances.

**Point to point response**

**Comment 1: Introduction,** The title focused on two factors, namely labelling processes and sieving methods, to aggregate turnover and SOC. Insufficient statements on the importance of these two factors are provided, instead, authors illustrated more the interaction between SOC and soil structure.

**Reply1:** We gratefully appreciate for your valuable comment. In the Introduction section, we have (1) deleted unnecessary description of the relationship between aggregate turnover and soil organic carbon dynamics; (2)Restructured introduction section from labeling and sieving processes on aggregate turnover and organic carbon dynamics, to make the introduction section more relevant to the research topic (**L25-L88**).

**L25-L88:** Soil structure is a crucial ecosystem service essential for maintaining physical, chemical, and biological processes in the soil. Soil aggregates, clusters of soil particles that adhere to soil organic components, are the fundamental units of soil structure. Soil aggregate dynamics involve aggregate formation, stabilization, and breakdown processes (Oades, 1991; Six et al., 2004). However, evidence shows that soil structure is never in a stable state but is constantly changing (Leij et al., 2002). Studies have suggested that this dynamic behavior mediates, to a large degree, soil organic matter (SOM) turnover and storage (Six et al., 1998; Plante and McGill, 2002). Although considerable studies have investigated soil aggregate dynamics, the life cycle of an aggregate and its impact on microbial-mediated C cycling remain elusive. Hence, quantifying soil aggregate turnover and SOM dynamics is essential to advance the understanding of SOM dynamics (Plante et al., 2002; De Gryze et al., 2006; Peng et al., 2017) and provide a better prediction of the response of the soil system to management practices.

Over the past 40 years, few studies have monitored soil aggregate dynamics using tracers. Indeed, stable isotopes (e.g., 137Cs, 210Pb, 7Be, and 234Th) were first used as tracers to monitor soil aggregate dynamics (Brown, 1981; Ritchie, 1990; Wallbrink, 1993). Then, Olmez (1994) proposed a soil particle labeling method for the diffusion of Au and Ag into sediment crystal lattice by high-temperature heating. However, heating may greatly change the chemical composition of the soil, especially soil clay and soil organic matter, making this method disadvantageous. Others like Staricka et al. (1992) mixed 1-3 mm ceramic spheres into the soil as tracers and found that they incorporated much more rapidly into macroaggregates (>40 mm) than into microaggregates (12-40 mm). However, All these methods have different drawbacks in labeling soil aggregates. Therefore, rare earth element oxides (REE oxides) were proposed to monitor the dynamics of soil aggregates, as they are harmless to the environment and characterized by small particle size (<5 μm), similar

physicochemical properties, low background concentrations, limited mobility, and low solubility, resulting in high sensitivity analysis, convenient monitoring results, and low-cost measurement and providing excellent properties as a potential mineral tracer (Tyler, 2004; De Gryze et al., 2006; Peng et al., 2017).

As mentioned above, the REE oxides strongly bind with silty and, in particular, clay soil particles are incorporated into soil aggregates (Zhang et al., 2001), explaining the widespread use of REE oxides as tracers of soil aggregates. Indeed, To obtain reliable results, soil aggregate tracers must be as unaffected and exhibit uniform states in soil. De Gryze et al. (2006) used the mean weight diameter (MWD) to determine the effect of REE oxides addition and labeling processes on soil aggregate,showed no evidence that the tracer affected microbial activity or initial aggregate size distribution. At the same time, calculated the rare-earth element (REE) enrichment percentages in each aggregate size,found the coefficients of variation of the rare-earth element measurements in eight spatially separate samples were acceptably small and the recovery rates large. From this, Liu et al. (2019) showed no evidence of potential effects of REE oxide types and addition on aggregate recovery rate and aggregate distribution in different soil types. Meanwhile, Morris et al.(2019) provided qualitative snapshots of eight REE oxides labeled aggregate thin sections through the X-ray fluorescence data, which also consistent with the results reported in the previous studies. The above studies demonstrate that REE oxides addition and labeling processes have little effect on soil aggregates, but few studies have concerned the effect of REE oxides addition and labeling processes on soil organic carbon dynamics.

Soil total organic carbon (TOC), occluded organic carbon(oPOC) and heavy fraction(HF) are regarded as slow fractions of soil organic carbon because they change slowly over time because of their large contents, protected by aggregates and associated with minerals (Franzluebbers et al., 1999;Riggs et al.,2015;Marín-Spiotta et al., 2008). In contrast, dissolved organic carbon (DOC) and microbial biomass carbon are considered as active organic carbon fractions that change seasonally (Wilson and Xenopoulos, 2008; Babur and Dindaroglu, 2020) and influence aggregate turnover(Murugan et al., 2019; Bucka et al., 2019). Similarly, free particulate organic carbon (POC) are considered as intermediate organic carbon fractions for changes of soil organic carbon with time that provide substrates for microbes (Witzgall et al., 2021) and also influence aggregation (Bucka et al., 2019). Because the aggregate distribution and aggregate turnover influence soil organic carbon (Six et al., 1998), determination of soil organic fractions provides important information on soil organic carbon sequestration and mineralization with aggregate turnover(Qiu et al., 2023).

The soil aggregate and associated carbon (C) fractions, however, are also influenced by the sieving method (Kemper and Rosenau, 1986; Whalen and Chang, 2002). The most common method of separating aggregates from bulk soil is the wet sieving

method, and the wet sieving method is widely used to determine size distribution and stability of aggregates caused by raindrop impact on dry soil causing slaking and the gas pressure inside aggregates (Elliott, 1986; Cambardella and Elliott, 1993). Compared with wet sieving method, The dry sieving method is based on the mechanical impact on soil structure during shaking the samples in a sieve tower, either by hand (Bach & Hofmockel, 2014) or using a mechanical sieve shaker(Nahidan & Nourbakhsh, 2018). As a result, dry sieving method for determining microbial biomass and activities that include water-soluble C and N in aggregates is receiving increased attention because of less destruction to the physical habitat of microbial communities(Kooch et al., 2022). Although information on soil organic carbon fractions is available (Kemper and Rosenau,1986), little is known about variations in soil organic carbon fractions with aggregate size fractions separated by dry and wet sieving methods.

On the other hand, REE oxides have been used in limited soil types, such as Alfisol in the studies of De Gryze et al. (2006) and Morris et al. (2019); Ultisols, Mollisols, and Inceptisols in the studies of Liu et al. (2019); and Vertisols in the study of Rahman et al. (2019). Andisols, the major soil type in Japan, are characterized by high contents of short-range-order (SRO) minerals and organometal complexes, very low bulk density, and a physically stable aggregate structure (Takahashi et al. 2016). A lack of study still exists on the use of REE oxides as tracers to label Andisols and the effect of the labeling process on Andisols organic carbon dynamics. Therefore, according to Peng et al. (2019), the study was divided into the labeling - sieving stage and the Recombination - incubation stage, for both short- and long-term effects, aims to assess (1) the effect of the labeling and sieving processes on soil aggregate turnover, (2) the effect of the labeling and sieving processes on soil organic carbon fractions , and (3) the impact on the quantitative analysis of the relationship between organic carbon and aggregate dynamics. Unlike previous studies that have focused on the effects of REE oxides labeling and sieving methods on soil aggregate size fractions, this paper provides the perspective of soil organic carbon fraction for the feasibility of using REE oxides as Andisols aggregate tracers.

**Comment 2:** L80 More details on the investigated soil should be provided, such as the initial SOC content, sand/clay/silt content, bulk density etc.

**Reply2:** We gratefully appreciate for your valuable comment. We provide the the soil properties in 2.1Soil characteristics in revised manuscript ( *2.1 Soil characteristics,* L98-L101).

**L98-L101:** Prior to the labeling and incubation experiment, the soil properties were: bulk density:0.78g cm$^{-3}$; soil pH: 6.50; total organic carbon (TOC): 53.21 mg g$^{-1}$, total N: 4.54 mg g$^{-1}$ , short range order mineral(SRO: allophane + ferrihydrite): 168 mg g$^{-1}$; Sand (2.0–0.05 mm): 23.92%; Silt (0.05–0.002 mm): 31.02%; Clay (<0.002 mm):

45.06%.

**Comment 3:** L90-100 Four oxides were used for labelling, but only 3 aggregate fractions were used? So which three oxides you used herein? How to get the recombined soil columns? The soil content and bulk density of these recombined columns? How many soil columns in total? More detailed information is needed.

L120 What do you mean by "regularly"? Every two days?

**Reply3:** We feel sorry for the inconvenience brought to the reviewer.

(1) We have placed the labeling process in the *2.3 Experiment design* section, together with sieving process, recombination process and incubation process. And the labeling process is described in *2.3.1 Labeling process* (**L127-L131**).

**L127-L131:** The REE oxides labeling process has been described elsewhere (Peng et al., 2017). Briefly, each REE oxides was suspended in ultra pure water by vortex mixing at a concentration of 300 mg kg$^{-1}$. 50g soil was continuously mixed and sprayed slowly with the REE oxides suspensions to homogenize labeling, the labeled soil was then stored at 4°C for 7 days to allow water equilibration with minimal microbial activity, followed by incubation at 25°C for 7 days. Next, the soil was oven-dried at 40°C for 48 h and broken down by hand to pass through a 2-mm sieve.

(2) For details of the recombination process is described as *2.3.3 The recombination process* and the protocol for recombining aggregates into recombined soils was added, as shown in *Figure 1* (**L143-L153**).

**L143-L153:** The recombination process includes the recombination of REE oxides labeled aggregates and the recombination of blank aggregates. For the REO-D and REO-W treatments, the recombined soil consisted of the 2-0.25 mm fraction labeled by Gd2O3 (A), 0.25-0.053 mm fraction labeled by Sm2O3 (B), and <0.053 mm fraction labeled by Nd2O3 (C). The protocol for combining REE oxides labeled aggregates into recombined soil is illustrated in Figure 1a. And for the CK-D and CK-W treatments, the recombined soil consisted of the 2-0.25 mm fraction(A), 0.25-0.053 mm fraction(B), and <0.053 mm fraction(C) without REE oxides. The protocol for combining aggregates without REE oxides into recombined soil as shown in                                                  Figure                                                  1b.

[Figure]

| | |
|---|---|
| a)The flow chart of the soil recombined by REE oxides labeled three different aggregate fractions. | b)The flow chart of the soil recombined by aggregate fractions without REE oxides. |

**Figure 1:The protocol for combining aggregates into recombined soil. A, B and C indicate 0.25-2 mm, 0.053-0.25 mm, and<0.0.53 mm aggregates, respectively.**

(3)Details of the container for recombined soil and the way of maintaining soil moisture content were described in *2.3.4 Incubation process*(**L155-L161**).

**L155-L161:**Five different treatments samples were gently packed into a PVC box (20 cm×20 cm×5 cm) with a flat platen to achieve a bulk density of 0.78g cm−3. The moisture content was maintained at 60% water-holding capacity for the whole incubation process by weighing every two days and supplying the lost water using a dropper. The aggregate turnover and SOC fractions were measured by destructively harvesting batches of soil on 0, 7, 14, 21, and 28 days of incubation. On the one hand, the feasibility of using REE oxides as tracers for aggregate turnover process was verified, and, on the other hand, The short- and long-term effects of the REE oxides labeling and sieving process on soil organic carbon were analyzed. All treatments were replicated three times.

**Comment 4:**L125 How much soil is used for dry/wet sieving and SOC fraction detection, respectively?

**Reply4:** Thank you for your rigorous consideration. We would like to respond this comment from two parts:(1) sieving methods; and (2) soil organic carbon fractions.

(1) **For the dry/sieving aggregate fraction.** Because five treatments were designed in this experiment, we prepared 500g soil samples for each treatment, except for the BG treatment, the treatments involved the labeling process. After labeling process, all labeled soil samples were sieved and recombination, 50 g of sample was sieved at each time (as shown in *2.3.2 Sieving process*).

  1) Where the descriptions of five treatments are described in **L114-L125**

**L114-L125:** Five treatments were designed as follows: (1) soil without REE oxides labeling and sieving processes (background treatment, BG), (2) soil with dry sieving and REE oxides labeling (REO-labeled and dry sieved treatment, REO-D), (3) soil with wet sieving and REE oxides labeling (REO-labeled and wet sieved treatment, REO-W), (4) soil with dry sieving but without REE oxides labeling (dry sieved treatment, CK-D), and (5) soil with wet sieving but without REE oxides labeling (wet sieved treatment, CK-W). Where (1) soil without REE oxides labeling and sieving processes (background treatment, BG) is a soil sample that passed a 2 mm sieve, served as a control without any processes. and the other treatments were subjected to labeling, sieving and recombination processes.In (4) and (5), the soil samples were subjected to the labeling process without the addition of REE oxides, followed by dry and wet sieving to obtain dry-sieving aggregates and wet-sieving aggregates,and finally recombined as CK-D treatment and CK-W treatment,which served as a control without REE oxides addition. For (2) and (3), the REE oxides labeled soil samples were sieved dry and wet to obtain REE oxides labeled dry sieving aggregates and REE oxides labeled dry sieving aggregates, which were finally recombined as REO-D treatment and REO-W treatment. Each treatment replicated three times.

  2) Where the descriptions of sieving process are described in **L133-L141**

**L133-L141:** The soil sample measuring <2 mm was separated into three fractions using both wet and dry sieving methods. The wet sieving method was performed as described in a previous study (Elliott, 1986). Briefly, 50g soil samples were placed on a filter paper in a mesh, and distilled water was added along with the edge of the filter paper until the soil sample was saturated. The soil sample was then placed into the top sieve of each set and rapidly immersed in distilled water while being oscillated for 2 min at a displacement of ~3.00 cm at 30 rounds per minute. Water-stable aggregates of 2.00 – 0.25, 0.25 – 0.053, and >0.053 mm were obtained. After sieving, the soil aggregates of each size were collected, oven-dried at 40℃, and weighed.

The dry sieving method was performed according to Sainju et al. (2003). Briefly, 50g soil samples were placed on the sieve, the sieve group was set according to the pore size, and the sieve was shaken by hand for 10 min. Three different dry sieve soil aggregates measuring 2 – 0.25, 0.25 – 0.053, and >0.053 mm were separated.

(2) **For the soil organic carbon fractions.** Different weights of soil samples were selected according to soil organic carbon fraction determination methods. Where,Total organic carbon(TOC): 0.15 – 0.20 g; Microbial biomass carbon(MBC):20.00g; Dissolved organic carbon (DOC): 20.00g; free particulate organic carbon (fPOC), occluded particulate organic carbon (oPOC), and a heavy fraction (HF):4.00g. These details about weight have been supplemented in *2.4.2 Analysis of soil organic carbon fractions*(**L173-L207**).

**Comment 5:**L175 How to calculate the aggregate turnover?

**Reply5:** We feel sorry for the inconvenience brought to the reviewer. The previous manuscript placed the calculation process in the appendix, which may have caused inconvenience to readers. Therefore, in revised manuscript, we have (1) introduced the calculation procedure in *2.5.1 Calculation of soil aggregate turnover*; (2) inserted the schematic diagram of aggregate turnover, to make it easier for readers to understand(**L209-L250**).

**L209-L250:** The mathematical description of aggregate turnover is considered a multidimensional extension of a simple first-order linear compartment model (De Gryze et al., 2006). Briefly, the current study tracked three different aggregate fractions, viz., A (0.25- to 2-mm fraction), B (0.053- to 0.25-mm fraction), and C (<0.053-mm fraction). The three aggregate fractions can produce three breakdown pathways from larger aggregate fractions into smaller aggregate fractions (a–c) and three buildup pathways from smaller into larger aggregate fractions (d–f). The mass transfer and turnover rate for each aggregate fraction were modified following the method in Peng et al. (2017).

[Figure]

Figure 2: The 6 possible transformation pathways of aggregate buildup (a – c) and breakdown (d – e) process among the three aggregates fraction. A, B, and C represent 0.25 – 2 mm, 0.053 – 0.25 mm and < 0.053 mm aggregates, respectively.

The mass of soil transfers along each pathway from time t1 to t2 can be described

with a discrete transfer matrix K(t2-t1)(Peng et al., 2017):

$$K_{(t_2-t_1)} = \begin{bmatrix} 1-a-b & d & f \\ a & 1-c-d & e \\ b & c & 1-e-f \end{bmatrix} \tag{1}$$

where a to l are the changes of proportions of REE oxides relating the specific pathways in Figure 2 from time t1 to t2, which is equivalent to the changes of proportions of aggregates relating the specific pathways.

First, we gain the REE oxide concentrations of different aggregate fractions at time t1 as follows:

$$REO_{con.(t)} = \begin{bmatrix} [Gd_A] & [Sm_A] & [Nd_A] \\ [Gd_B] & [Sm_B] & [Nd_B] \\ [Gd_C] & [Sm_C] & [Nd_C] \end{bmatrix} \tag{2}$$

where, e.g., $Gd_A$ is the concentration of Gd in small macroaggregate (A,0.25-2.00 mm) fraction. The amount of aggregates at time steps t can be described by vectors S(t) :

$$S(t) = \begin{bmatrix} A(t) \\ B(t) \\ C(t) \end{bmatrix} \tag{3}$$

where the A, B and D represent the amounts of small macroaggregates (0.25-2 mm), microaggregates (0.053-0.25 mm), and silt and clay sized aggregates(<0.053 mm), respectively.

The absolute REE oxide amounts in the three aggregate fractions are:

$$REO_{amo.(t)} = \begin{bmatrix} A(t)[Gd_A] & A(t)[Sm_A] & A(t)[Nd_A] \\ B(t)[Gd_B] & B(t)[Sm_B] & B(t)[Nd_B] \\ C(t)[Gd_C] & C(t)[Sm_C] & C(t)[Nd_C] \end{bmatrix} \tag{4}$$

When the absolute tracer amounts in aggregates is assumed during transfer between time steps t1 and t2, their relationship can then be described as follows:

$$REO_{amo.(t2)} = K(t_2 - t_1)REO_{amo.(t1)} \tag{5}$$

Consequently, the transformation matrix K(t2-t1) can be calculated:

$$K(t_2 - t_1) = REO_{amo.(t1)}^{-1} REO_{amo.(t2)} \tag{6}$$

where the K(t2-t1) indicates the change in the proportions of aggregates falling into sizes A, B or C between time steps t1 and t2 and f indicates the pathway of C fraction buildup into A fraction, while b indicates the pathway of A fraction breakdown into C fraction (Figure 2).

Because the Andisols samples were subjected to the labeling process, the sieving process, and the recombination process, and finally to incubation, The labeling, sieving and recombination processes have a destructive effect on aggregates. Soil samples obtained from 7-, 14-, 21- and 28-days incubation included both the labeling-sieving and recombination processes and the incubation process($K_{tx}$), whereas samples from 0 day incubation included only the labeling, sieving and recombination processes ($K_{t0}$), then the contribution of the incubation effect to aggregate turnover is calculated as:

$$K_{inc} = K_{tx} - K_{t0} \tag{7}$$

Finally, the predicted aggregates fractions were calculated as:

$$A_{t2} = (1 - a - b)A(t_1) + dB(t_1) + fC(t_1) \tag{8}$$

$$B_{t2} = aA(t_1) + (1 - c - d)B(t_1) + eC(t_1) \tag{9}$$

$$C_{t2} = bA(t_1) + cB(t_1) + (1 - e - f)C(t_1) \tag{10}$$

**Results**

**Comment 6:** L190 Since the results and discussion parts are separated herein, no reference should be included in results part.

**Reply6:** We gratefully appreciate for your valuable comment. Following your suggestion, the discussion and references in *Results* section have been moved to *Discussion* section.

**Comment 7:** L200 Please explain the meaning of "unaffected carbon pools".

**Reply7:** Thank you so much for your careful check. In previous manuscripts, as oPOC and HF fractions in Table 1 were less affected by REE oxides addition, labeling, sieving and recombination processes, we have attempted to unify this part of the soil organic carbon fractions into 'unaffected carbon pools' for discussion. In revised manuscript, we did not add new concepts( like 'unaffected carbon pools') , but described all carbon fractions according to Table3(**L356-L359**).

**L346-L349:** The effects of the REE oxides addition, labeling process and sieving process on soil organic carbon were evaluated by assessing the differences in the organic carbon fractions of different treatments at the 0-day incubation. The TOC, oPOC and HF fractions were not significantly different between treatments (P>0.05). The TOC was maintained at 48442.64-50108.35 mg/g soil, oPOC fraction content at 2543.59-2665.78 mg/g soil and HF at   25439.00-36714.17 mg/g soil.

**Comment 8:**L210 It will be easier for readers to follow when 0.25- to 2-, 0.053- to 0.25-, and <0.053-mm are replaced by 0.25-2 mm, 0.053-0.25 mm and <0.053 mm.

**Reply 8:** We feel sorry for the inconvenience brought to the reviewer. Following your suggestion, we have replaced 0.25- to 2-, 0.053- to 0.25-, and <0.053-mm with 0.25-2 mm, 0.053-0.25 mm and <0.053 mm in revised manuscript.

**Comment 9:**L265 I doubt the calculation on aggregate turnover. Take turnover rate of 0.25-2 mm at 7 days as an example, 0.75=(4.58+0.68)/7, it seems that the formation processes are not taken into consideration, which is different from the calculation proposed by De Gryze et al. and Peng et al.

L270 According to Fig.5, the breakdown and formation of dry sieving aggregates occurred not only the first week.

L275 Transformation pathways in Fig. 6 are much smaller than published data. Why? No further discussion are displayed.

**Reply 9:** We totally understand the reviewer's concern. These three questions are about the transformation of aggregate turnover pathways, so we would like to provide better responses to your comments.

In earlier manuscripts, we were too concerned with the relationship between soil aggregate turnover and soil organic carbon, and therefore removed the transformation of aggregate turnover pathways before incubation(0 days) as a disturbance. Actually, to elucidate the influence of the labeling and sieving processes on the aggregates turnover, The transformation paths of three aggregate fractions were divided into (1) turnover directly caused by the labeling and sieving processes (at 0 days incubation); and (2) turnover caused by soil microorganisms during the incubation process (at 7,14,21,28 days incubation). Soil samples obtained from 7, 14, 21 and 28 days of incubation included both the labeling-sieving and recombination processes and the incubation process, whereas samples from 0-day incubation included only the labeling, sieving and recombination processes. Therefore, the turnover pathways of the incubation process are calculated as the difference between the turnover pathways of different incubation days (7, 14, 21, 28days) and the turnover pathways of 0 days of incubation.

In the revised manuscript, we have (1) introduced the labeling and sieving process and the incubation process in *2.3 Experimental design* (**L108-L113**);.

**L108-L113:** A series of experiments were conducted in this study. First, The feasibility of REE oxides as tracers to track Andisols aggregate turnover was determined. Then, we divide the effects of REE oxides on Andisols aggregate turnover and organic carbon dynamics into two processes: labeling and sieving processes and incubation process. In the labeling and sieving processes, REE oxides addition, labeling method and sieving method are the main causes of soil organic

carbon and aggregate turnover. And in the incubation process variations in soil organic carbon dynamics and aggregate turnover are caused by initial soil organic carbon fractions differences and the soil microbial.

(2) added *Equation(7)* in the calculation section as a supplement to the calculation procedures (**L241-L246**).

**L241-L246:** Because the Andisols samples were subjected to the labeling process, the sieving process, and the recombination process, and finally to incubation, The labeling, sieving and recombination processes have a destructive effect on aggregates. Soil samples obtained from 7-, 14-, 21- and 28-days incubation included both the labeling-sieving and recombination processes and the incubation process(Ktx), whereas samples from 0 day incubation included only the labeling, sieving and recombination processes (Kt0), then the contribution of the incubation effect to aggregate turnover is calculated as:

$$K_{\mathrm{inc}} = K_{tx} - K_{t0} \tag{7}$$

(3)included images and analysis of the transfer pathways between the three aggregate size fractions before incubation(0day) in *3.1.2 Soil aggregate turnover pathways* (**L291-L314**).

**L291-L314:** In analogy to soil organic carbon dynamics, the labeling and sieving processes were also divided into the labeling-sieving processes and incubation process for soil aggregate turnover pathways. Where, Before incubation(0day), the recombination and sieving effect caused the aggregates to fragment and recombine. The transfer pathways between the three aggregate size fractions caused by the labeling and sieving processes are shown in Figures 5.

[Figure]

a) BG treatment                                                    b) Dry-sieving treatment

[Figure]

c)  Wet-sieving treatment

**Figure 5. The six transformation pathways of three dry sieving aggregate fractions on 0 day of incubation. Values in *arrows* are the relative changes in aggregate fractions (in percentage). *A*, *B*, and *C* represent the 0.25-2 , 0.053-0.25 , and <0.053 mm Andisols aggregate fractions, respectively.**

Ideally, the labeling and sieving process would not affect the soil aggregates, and therefore soil aggregate turnover would not occur before incubation (Fig. 5a). By comparing aggregate turnover between two sieving treatments and BG treatment, it was found that the labeling and sieving methods had a significant effect on Andisols aggregates turnover($p<0.05$), and the effect on the aggregates turnover were different depending on the sieving methods. The labeling and dry sieving process resulted in 28.19% of the 0.25-2mm aggregate fraction breaking down, with 14.51% breaking down to 0.053-0.25mm aggregate fraction, and 13.58% breaking down to <0.053mm aggregate fraction. Meanwhile, 17.83% of the 0.053-0.25mm aggregate fraction formed the 0.25-2mm aggregate fraction in this process, and 11.56% fragmented to the 0.053mm aggregate fraction. Whereas <0.053mmfraction was most affected by the labeling and sieving process, 33.71% of this fraction formed the 0.25-2mm aggregate fraction and 30.79% formed the 0.053-0.25mm aggregate fraction. Unlike the dry sieving treatment, the wet sieving treatment was more affected by the labeling and sieving process. In particular, the turnover between 0.053-0.25mm aggregate fraction and <0.053 mm aggregate fraction, where 24.71% of the 0.053-0.25mm aggregate fraction was break down to <0.053mm aggregate fraction , while 38.25% of the <0.053mm aggregate fraction aggregated to form 0.053-0.25mm aggregate fraction.

(4)Compared with Peng et al. (2017) and M. Halder et al. (2022) for transformation of aggregate turnover pathways and turnover rates in Discussion section *4.1Effects of labeling and sieving processes on Andisols aggregate* (**L445-L488**).

**L445-L488**: To elucidate the influence of the labeling and sieving processes on the aggregates turnover, The transformation paths of three aggregate fractions was divided into: (1) turnover directly caused by the labeling and sieving processes (at 0 days incubation); and (2) turnover caused by soil microorganisms during the incubation process (at 7,14,21,28 days incubation). Labeling and sieving processes affect the aggregate turnover pathways before incubation(0 day). In particular, the turnover pathway of wet sieving aggregates (Figure 8 and 10) is consistent with the study in Peng et al. (2017), where at 0 days incubation, soil aggregates were mainly affected by labeling, sieving and recombination processes, producing a large number of aggregate breakdown-formation transformation pathways. In the study by Peng et al. (2017), the 0.25-2mm wet sieving aggregate fraction was used as an example. 26% broken down to 0.053-0.25 aggregate fraction and another 26% broken down to 0.053% aggregate fraction. The effect of wet sieving on soil aggregates has been shown to depend on size, structure, shrinkage and expansion behavior, wettability of soil components, porosity and the spatially heterogeneous distribution of these properties within the aggregates (Baumgartl & Horn, 1993; Chenu, Le Bissonnais, & Arrouays, 2000; Kaiser, Kleber , & Berhe, 2015). De Gryze et al. (2006) used artificial wet sieve agglomerates and after the sieving and recombination process, 30% of the 0.25-2mm aggregate fraction broken down to 0.053-0.25 aggregate fraction, and 31% broken down to <0.053mm aggregate fraction, this fraction higher than the transformation of turnover pathways in our study;Another reason might be the different fractions of aggregates. In previous studies, 2-5mm fraction, 0.25-2mm fraction, 0.053-0.25mm fraction and 0.053mm fraction were studied as simplifications of soil structure (Liu et al.,2019), and since Andisols has a small proportion of 2-5mm fractions (Asano et al.,2014). Therefore, in this study, three aggregate fractions (0.25-2mm fraction, 0.053-0.25mm fraction and 0.053mm fraction)were selected. The effect of labeling and sieving on dry sieving aggregates has not been studied before. Morris demonstrated the effect of AMF on aggregates turnover by dry sieving aggregates turnover, but the study only showed the aggregates turnover path after 5 week incubation and no further study of the effect of the labeling and sieving process was performed. The comparison of sieving methods demonstrated that although wet-sieving aggregate fractions were found to contain more free primary particles and fine-sized aggregates, dry-sieving aggregate fractions contained more aggregates with inclusions of primary particles.

Similarly, the labeling and sieving processes also impact on aggregate turnover pathways during the incubation process. Since the turnover path is the sum of the labeling and sieving effects and the incubation process, the turnover pathways of the incubation process is calculated as the difference between the turnover pathways of different incubation days and the turnover pathways of 0 days incubation (as equation 7). Due to the different topics of the studies, Peng et al. (2017), De Gryze et al. (2006)

and M. Halder et al (2022) did not distinguish the wet sieve aggregate turnover pathways. With the same approach to divide the aggregate turnover pathways in Peng et al (2017), With the same approach to divide the aggregate turnover pathways in Peng et al (2017), it was found that during the incubation period, the aggregate turnover mainly occurred in 0-7days, with 16% of the 0.25-2 mm aggregate fraction was broken down to 0.053-0.25 mm aggregate fraction and 13% to 0.053 aggregate fraction. In the following 7-14 days, the proportion of 0.25-2mm aggregate fraction breaking down to 0.053-0.25mm aggregate fraction was -2%, and the proportion breaking down to 0.053 aggregate fraction increased only 1%. Such trends are consistent with our study, and also consistent with the M. Halder et al's (2022) calculations for wet sieving aggregate turnover. In M.Halder et al.'s(2022) study,

The transformation pathways of 0.25-2 mm aggregate fraction breaking down to 0.053-0.25 mm aggregate fraction was 3%, which was similar to the breaking ratio of 0.25-2 mm aggregate fraction to 0.053-0.25 mm aggregate fraction in this experiment. For the effect of labeling and sieving processes on dry sieving aggregates turnover processes, Morris et al. (2017) classified dry sieving aggregates as 1mm ,0.212mm,0.053m and <0053mm aggregate fractions, and the experiment only reported aggregates turnover paths after 5 weeks of incubation, making it difficult to make comparisons.

The aggregates turnover rate during the incubation is also affected by the labeling and sieving process. The turnover rate of wet sieving aggregates showed the maximum rate from 0-7 days and then gradually decreasing. This is consistent with Peng et al (2017),where the 0.25-2 mm fraction showed turnover rates of 4.14%/day at 7 days, then decreasing to 2%/day at 14 days and further decreasing to 1.04%/day at 28 days during incubation. M. Halder et al. (2022) also showed the same trend that the wet sieving aggregates turnover faster at the beginning of the incubation period. The turnover rate of the 0.25-2 mm fraction wet sieving aggregates was 0.14%/day at 14 days, decreasing to 0.035%/day at 28 days of incubation.

**Comment 10:** L300 There are two "Wet-MBC" in Fig.7a? To present the same SOC fraction, authors used the same color in a, while used the same shape in b, please keep them uniformed.

**Reply10:** Thank you so much for your careful check. We apologize for our carelessness. In earlier manuscript, the grey circle represents the "Wet-DOC" instead of "Wet-MBC" in Fig. 7a. In the revised manuscript we have (1) corrected the error in the figure legend; (2) increased the dynamic of the BG treatment organic carbon fractions during incubation(*3.2.2 The effect of labeling and sieving processes on SOC fractions during incubation processes*, **L382-L404**).

**L382-L404:**As REE oxides addition had no significant effect on the initial state of the soil organic carbon pool, the REO-W and CK-W treatments were combined into the

wet sieving treatment, and the REO-D and CK-D treatments were combined into the dry sieving treatment when comparing long-term effects. The proportions of SOC fractions at different incubation days are shown in Figure 8.

[Figure]

**a** MBC fraction

**b** DOC fraction

**c** fPOC and oPOC   fractions

**d** HF fraction

Figure 8: The proportion of different carbon fractions in recombined soils with incubation time.

The differences in soil organic carbon during incubation caused by labeling and sieving processes were developed by comparing the treatments with the BG treatment. Where, DOC and MBC, the small fractions of SOC, remained stable in dry sieving recombined soil at 0.06%－0.08% compared with BG treatment during incubation. However, in wet sieving aggregate recombined soil, MBC and DOC showed different trends, with DOC first increasing from 0.09% to 0.11% in the first 7 days and then slowly decreasing to 0.07% at 28 days. The MBC decreased to 0.04% and remained in the range of 0.03%－0.04% from 7 to 28 days.

oPOC, which is believed to be protected by aggregates, exhibited an increasing trend

throughout the incubation period. The oPOC in dry aggregate recombined soil increased from 5.05% to 6.21% during 28 days incubation, as well as oPOC in wet aggregate recombined soil, increased from 5.50% to 6.00%. The fPOC fraction in dry sieving recombined soil increased from 1.97% to 2.70% through the first 7 days and then slowly decreased to 2.03% from 7 to 28 days. In contrast, for the wet sieving recombined soil, fPOC showed a high growth rate from 0 to 7 days as the fPOC was removed during the wet sieving process, followed by a slow increase to 4.44% on days 7-28. The HF showed remarkable similarity in both recombined soils, declining slowly in the first 21 days and then decreasing sharply after 21 days, Even though HF did not show a clear trend in the BG treatment.

**Discussion**

**Comment 11:**There are lots of repetition of results. No highlights were proposed and discussed here. For section 4.2, lots of publications have proved it, there is no need to discuss again. For section 4.4, the relationship between SOC and aggregate are analyzed, which should be displayed in results rather than discussion part.

**Reply11:**We feel sorry for the inconvenience brought to the reviewer. We have tried too much to illustrate the feasibility of using soil organic carbon fractions and REE oxides to quantify soil organic carbon dynamics and soil aggregate turnover, respectively, and to analyze their relationship. This resulted in the Discussion section being inconsistent with the research topic.

(1)We discussed the effect of labeling and sieving processes on the aggregate turnover from 1)the feasibility of REE oxides as Andisols aggregate tracers; 2) The transformation of aggregate turnover pathways before incubation (0d); 3) The transformation of aggregate turnover pathways during incubation;4) the aggregate turnover rate during incubation(*4.1 Effects of labeling and sieving processes on Andisols aggregate,* L436-L488).

**L436-L488:** Labeling and sieving processes have no effect on the feasibility of REE oxides as tracers to track aggregate turnover. The relationship between the measured and predicted results of soil dry sieving aggregates (Figure 3) was consistent with the finding reported by Morris et al.(2019). In the study of Morris et al.(2019), X-ray fluorescence data were used to provide insight into mechanisms operating during aggregate turnover, showing that REE oxides are effective tracers for studying the dry sieving aggregate formation and decomposition dynamics. Also, the close 1:1 line between measured and predicted aggregates (Figure 4) supported Peng et al.'s (2017) findings that REE oxides are effective tracers of wet sieving aggregate dynamics. Whilst REO was an effective tracer for both dry sieving and wet sieving aggregate, there were limitations. The weaker relationship for 0.053-0.25 mm(slope = 0.89) wet sieving aggregates (Figure 4) may be an experimental artefact due to the additional loss or gain of these fraction during the wet sieving procedure.

To elucidate the influence of the labeling and sieving processes on the aggregates turnover, The transformation paths of three aggregate fractions was divided into: (1) turnover directly caused by the labeling and sieving processes (at 0 days incubation); and (2) turnover caused by soil microorganisms during the incubation process (at 7,14,21,28 days incubation). Labeling and sieving processes affect the aggregate turnover pathways before incubation(0 day). In particular, the turnover pathway of wet sieving aggregates (Figure 8 and 10) is consistent with the study in Peng et al. (2017), where at 0 days incubation, soil aggregates were mainly affected by labeling, sieving and recombination processes, producing a large number of aggregate breakdown-formation transformation pathways. In the study by Peng et al. (2017), the 0.25-2mm wet sieving aggregate fraction was used as an example. 26% broken down to 0.053-0.25 aggregate fraction and another 26% broken down to 0.053% aggregate fraction. The effect of wet sieving on soil aggregates has been shown to depend on size, structure, shrinkage and expansion behavior, wettability of soil components, porosity and the spatially heterogeneous distribution of these properties within the aggregates (Baumgartl & Horn, 1993; Chenu et al., 2000; Kaiser et al., 2015). De Gryze et al. (2006) used artificial wet sieve agglomerates and after the sieving and recombination process, 30% of the 0.25-2mm aggregate fraction broken down to 0.053-0.25 aggregate fraction, and 31% broken down to <0.053mm aggregate fraction, this fraction higher than the transformation of turnover pathways in our study ;
Another reason might be the different fractions of aggregates. In previous studies, 2-5mm fraction, 0.25-2mm fraction, 0.053-0.25mm fraction and 0.053mm fraction were studied as simplifications of soil structure (Liu et al.,2019), and since Andisols has a small proportion of 2-5mm fractions (Asano et al.,2014). Therefore, in this study, three aggregate fractions (0.25-2mm fraction, 0.053-0.25mm fraction and 0.053mm fraction)were selected. The effect of labeling and sieving on dry sieving aggregates has not been studied before. Morris demonstrated the effect of AMF on aggregates turnover by dry sieving aggregates turnover, but the study only showed the aggregates turnover path after 5 week incubation and no further study of the effect of the labeling and sieving process was performed. The comparison of sieving methods demonstrated that although wet-sieving aggregate fractions were found to contain more free primary particles and fine-sized aggregates, dry-sieving aggregate fractions contained more aggregates with inclusions of primary particles.

Similarly, the labeling and sieving processes also impact on aggregate turnover pathways during the incubation process. Since the turnover path is the sum of the labeling and sieving effects and the incubation process, the turnover pathways of the incubation process is calculated as the difference between the turnover pathways of different incubation days and the turnover pathways of 0 days incubation (as equation 7). Due to the different topics of the studies, Peng et al. (2017), De Gryze et al. (2006) and M. Halder et al (2022) did not distinguish the wet sieve aggregate turnover

pathways. With the same approach to divide the aggregate turnover pathways in Peng et al (2017), With the same approach to divide the aggregate turnover pathways in Peng et al (2017), it was found that during the incubation period, the aggregate turnover mainly occurred in 0-7days, with 16% of the 0.25-2 mm aggregate fraction was broken down to 0.053-0.25 mm aggregate fraction and 13% to 0.053 aggregate fraction. In the following 7-14 days, the proportion of 0.25-2mm aggregate fraction breaking down to 0.053-0.25mm aggregate fraction was -2%, and the proportion breaking down to 0.053 aggregate fraction increased only 1%. Such trends are consistent with our study, and also consistent with the M. Halder et al's (2022) calculations for wet sieving aggregate turnover. In M.Halder et al.'s(2022) study,

The transformation pathways of 0.25-2 mm aggregate fraction breaking down to 0.053-0.25 mm aggregate fraction was 3%, which was similar to the breaking ratio of 0.25-2 mm aggregate fraction to 0.053-0.25 mm aggregate fraction in this experiment. For the effect of labeling and sieving processes on dry sieving aggregates turnover processes, Morris et al. (2017) classified dry sieving aggregates as 1mm ,0.212mm,0.053m and <0053mm aggregate fractions, and the experiment only reported aggregates turnover paths after 5 weeks of incubation, making it difficult to make comparisons.

The aggregates turnover rate during the incubation is also affected by the labeling and sieving process. The turnover rate of wet sieving aggregates showed the maximum rate from 0-7 days and then gradually decreasing. This is consistent with peng et al (2017),where the 0.25-2 mm fraction showed turnover rates of 4.14%/day at 7 days, then decreasing to 2%/day at 14 days and further decreasing to 1.04%/day at 28 days during incubation. M. Halder et al. (2022) also showed the same trend that the wet sieving aggregates turnover faster at the beginning of the incubation period. The turnover rate of the 0.25-2 mm fraction wet sieving aggregates was 0.14%/day at 14 days, decreasing to 0.035%/day at 28 days of incubation.

(2)We discussed the effects of the labeling and sieving process on soil organic carbon in terms of 1) the effect of wet sieving on soil organic carbon dynamics; 2) the effect of dry sieving on soil organic carbon dynamics; 3) the feasibility of using soil organic carbon fractions to analyze the relationship between organic carbon dynamics and aggregate turnover. (*4.2 Effects of REE oxide labeling and sieving processes on soil organic carbon*, **L490-L530**)

**L490-L530:**The addition of REE oxides has no effect on soil organic carbon dynamics, which is supported by Zhang et al. (2001), Peng et al. (2017), De Gryzel, and also REE oxides are considered have no effect on soil microorganisms. The effect of labeling and sieving on soil organic carbon is discussed in terms of both (1) the labeling and sieving processes and (2) the incubation process. The initial low content of the fPOC fraction in wet sieving aggregates(Table3) could be explained from two

aspects. On the one hand there are several wet-dry cycles during the labeling process, where dry-wet cycles are believed to consume POC (Denef et al., 2001); and on the other hand there is a significant amount of free organic particles (i.e. particles >53 μ m that are not enclosed within the aggregates) loss during the wet sieving process ((Haynes, 2000). And during the incubation process, the wet sieving aggregate turnover was found affected by soil microorganisms (Blaud et al., 2017) and the rewetting of dry samples during wet sieving was shown to affect the characterization of microbial communities in soil aggregates (Bach et al., 2018),It is consistent with our research that the proportion of the MBC fraction in soil organic carbon shows a decreasing trend with incubation time. In contrast to the MBC fraction, the fPOC fraction in soil organic carbon gradually increased, with other fractions decreased(as shown in Figure 8). These can be explained by(as reviewed by Kaiser et al., 2015) the formation of new interactions between minerals and OM, the killing of microorganisms and shifts in their community structure and an increase of mineral surface acidity and hydrophobicity (possibly affecting aggregate stability; (Ibrahimi et al., 2019)).The degree to which these characteristics are altered by wetting and drying depends on various factors, such as soil depth (Kaiser et al., 2015), land use (Fierer & Schimel, 2003), texture and OM content (Albalasmeh & Ghezzehei, 2014), sodium and calcium cation configuration (Aquino et al., 2011), pH value (Kang & Xing, 2008) and drying speed (Kemper & Rosenau, 1986). All of these influences alone are difficult to quantify, which is why (repeated and/or unnecessary) wetting and drying should be avoided to preserve the natural properties of soil aggregates and to prevent the creation of artifacts.

  The Dry sieving method avoids the changes in physical, chemical and biological properties that can be observed during drying and wetting(Sainju et al.,2003). This is also demonstrated in this experiment, the labeling and dry sieving processes also affect the soil organic carbon fractions, but the effects appeared smaller. In contrast to the wet sieving treatment.The fPOC and DOC fractions increased from 0-7 days with dry sieving treatment(Figures 8b&8c), which result partially from the decay of some soil organisms after drying (Lund and Goksoyr., 1980; Bottner, 1985) and partially from the release of previously inaccessible organic compounds (Sorensen, 1974; Van Gestel et al., 1991). The increased availability of organic substrates leads to increased microbial activity and increased growth of fungal mycelium and bacterial biomass (Jager and Bruins, 1974), this would explain the increased MBC content of the dry sieving treatment form 0-7days, and it remains stable at around 0.75%SOC (Figure 8a). For this reason, Bach & Hofmockel (2014) indicated that wet sieving process should be used to assess long-term changes in microbial activity and organic carbon fractions, while dry sieving process without rewetting could captured short-term dynamics of soil organic carbon as well. Thus, the effects of labeling and sieving processes on aggregates and organic carbon need to be considered, before using soil

organic carbon fractions and REE oxides to quantify organic carbon dynamics and aggregate turnover respectively.

Further data are needed to support the application of soil organic carbon fractions to quantify soil organic carbon dynamic, with REE oxides to track the aggregates turnover. In previous studies, Peng et al. (2017) analysed the organic carbon dynamics with by adding 13C-labeled glucose to REE oxides labeled soils and determining the 13C content in different aggregate fractions. Subsequently, M. Halder et al. (2022) used eleven organic materials were characterized in terms of nutrient stoichiometry, biochemical features and carbon (C) functional groups, to determine which characteristics of organic materials control soil aggregate turnover. In this study, the correlation between the aggregates turnover and organic carbon fraction dynamics was analyzed, and we found that fPOC was significantly correlated with the dry sieving aggregate turnover rate during the 28 days incubation. However, for more complex conditions, more research is needed to determine whether REE oxides and soil organic carbon fraction could be used, to quantify the relationship between aggregate turnover and organic carbon dynamics.

**Conclusion**

**Comment 12:** It is abstract, not conclusion. The main findings/conclusions, rather than results, are supposed to be included here.

**Reply12:** Thank you for your valuable suggestion.

Conclusions based on the research topic and discussion were obtained from aggregate turnover and soil organic carbon dynamics, respectively. The addition of REE oxides would have no effect on the Andisols aggregate turnover and organic carbon dynamics, but the REE oxides labeling and sieving processes would have effects on soil aggregates and soil organic carbon(**L532-L548**).

**L532-L548:** It was shown that the addition of REE oxides would have no effect on the Andisols aggregate turnover and organic carbon dynamics, but that the REE oxides labeling and sieving processes would have effects on soil aggregates and soil organic carbon. The experiments were divided into labeling & sieving processes and incubation process according to the mechanism of aggregate turnover. During the labeling and sieving processes, dry sieving aggregates showed a lower transformation of aggregate turnover pathways before incubation (0 d) compared to wet sieving aggregates. Aggregate turnover mainly occurred in 0-7 days, followed by a gradual decrease of both turnover paths and rates in 7-28 days. However, the labeling and wet sieving process involved several wet-dry cycles that may remove the bacterial cells, resulting in a lower turnover path and rate of wet sieving microaggregates. It is clear that the choice of sieving method was reflected in the different processes of aggregates turnover, and this should be considered in future studies.We suggest to measure the aggregates turnover at 0 days to eliminate the turnover caused by the

labeling and sieving processes.

For organic carbon dynamics, the dry sieving treatment retained more MBC and fPOC fractions after the labeling and sieving processes. During the incubation process, the labeling and sieving methods also play a role in the organic carbon dynamics. During the incubation process, the labeling and sieving methods also play a role in the organic carbon dynamics. The organic carbon dynamics of the dry sieving process are more consistent with those non-labeled & sieving processes soils. As we only studied Andisols organic carbon dynamics and aggregate turnover without the addition of exotic organic carbon, further research is still needed to support the application of soil organic fractions and REE oxides tracers to quantify the relationship between the soil organic carbon dynamics and aggregate turnover.

**Acknowledgments**

We gratefully thanks for the precious time the reviewer spent making constructive remarks.

**L552-L553:** We also acknowledge one anonymous reviewer for helpful comments on an earlier draft of our manuscript.

**Reference**

Based on your suggestions, the Introduction and Discussion sections have been revised a lot, so we have restructured the Reference section.(8 Reference, **L557 - L672**).

We hope you will find our revised manuscript acceptable for publication.

Yours sincerely,

Wang Yike

---

## Author Comment (AC3)

We gratefully thank the editor and reviewer for the time spent making their constructive remarks and helpful suggestion, which has significantly raised the quality of the manuscript and has enabled us to improve the manuscript. Each suggested revision and comment, brought forward by the reviewer was accurately incorporated and considered. Below the reviewer's comments are response point by point and revisions are indicated.

**'Comment on egusphere-2022-728', Anonymous Referee #2, 22 Nov 2022**

**General Comments:** This manuscript by Wang et al. presented an important and interesting study of the effects of REOs labeling process and sieving methods on aggregate turnover and carbon dynamic. Researching soil aggregates and associated biogeochemical processes is a very time-consuming and laborious work, with the complex REOs labeling process makes it to be more difficult. Recently, studies focusing on aggregate turnover show a rising trend, while less researches have reported the role of labelling and sieving processes played in affecting aggregate turnover and soil C dynamics, which may obscure or even magnify the effect of treatments. Dividing the labeling process from the incubation (or other treatments) thus is pivotal for accurately assess the real aggregate turnover dynamics and soil C dynamics. Indeed, the authors found that labeling process and sieving method affect aggregate turnover and soil C fractions (particularly labile C fraction, i.e., DOC, MBC) more intensively than incubation. However, the text has low readability. The logic is very confusing, and the authors seem unable to catch the highlights and key points of the story. Additionally, some key information about the calculations do not show in article. I very appreciate with the work and its significance the authors done. However, I feel a pity that the authors show the story in a not good way. So, I do not recommend the publication of the article in SOIL. The best result is resubmission after major revision.

**Response:** We greatly appreciate the reviewer's insightful comments. In fact, while processing the data for this manuscript, we found that there was perhaps a hidden innovation, the application of soil organic fractions to quantify soil organic carbon, with REE oxides to track the aggregate turnover. In previous studies, Peng et al. (2017) analyzed the organic carbon dynamics by adding 13C-labeled glucose to REE oxides labeled soils and determining the 13C content in different aggregate fractions. Subsequently, M. Halder et al. (2022) used eleven organic materials characterized in terms of nutrient stoichiometry, biochemical features and carbon (C) functional groups, to determine which characteristics of organic materials control soil aggregate turnover. However, in our following studies, we found that it would be too expensive to use carbon isotope methods in field experiments or to determine the contribution of organic carbon monomers (e.g. galactosamine) for aggregate turnover. This is why

some parts focus on the effect of labeling and sieving processes to aggregate and SOM dynamics, and some parts focus on the descriptions of the relationship between aggregates and organic carbon in previous manuscripts, Which resulted in the manuscript's logic seems pretty confusing and not well readable. Apologies again to the reviewers.

Based on your suggestions, we have restructured the logical framework of the manuscript and will respond to your suggestions in a point-to-point response. In the revised version of the manuscript, we have refined the abstract and main text (especially the Introduction, the Results and the Discussion sections) to make the paper easier to read, the procedure for the calculation of aggregate turnover, which was originally placed in the appendix, has also been collated into 2.5.1 Calculation of soil aggregate turnover in the revised manuscript.

**In the Introduction section**, we have (1) restructured the framework of the manuscript to make the manuscript more palatable to general readers; (2) outlined the major assumptions briefly; (3) deleted unnecessary description of the relationship between aggregate turnover and soil organic carbon dynamics to make the introduction section more relevant to the topic.

**In the Material and method section,** we have (1) introduced a more specific description of the Andisols soil samples in *2.1 Soil characteristics*; (2)Changed the description of the experiment design in *2.3 Experiment design* to make it more consistent with the research topic. (3) Added a flow chart of the recombination process in *2.3.1 Recombination process*, to make the recombination process more accessible to the readers; (4) Added *2.5.1Calculation of soil aggregate turnover* in the revised manuscript, from the original appendix and added Figure2 The 6 possible transformation pathways of aggregate.

**In the Results section,** we have (1) modified the structure of the result section according to the revised experiment design, and described the effect of the labeling and sieving process on aggregate turnover and organic carbon fractions, respectively; (2)added transformation aggregates turnover pathways before incubation (0 days) in *3.1.2 Soil aggregate turnover pathways*; (3) Added soil organic carbon fraction dynamics of BG treatment during incubation in 3.2.2 The effect of labeling and sieving processes on SOC fractions during incubation process; (4) placed the relationship between aggregate turnover and organic carbon dynamics in *3.3 The effects on the quantitative study of the relationship between aggregate turnover and organic carbon dynamics*.

**In the Discussion section,** we have reorganized the discussion according to the research topic and your comments. the impact of the labeling and sieving processes on soil aggregate turnover and soil organic carbon fractions were discussed, respectively.

**In the Conclusion section,** We have (1) identified that labeling and sieving processes could affect aggregate turnover and soil organic carbon fractions; (2) made

suggestions for eliminating the disturbances.

**Point to point response**

**Comment 1: Introduction,** The introduction does not align well with the topic. The article aims to reveal the effects of labeling process and sieving method on aggregate turnover and soil C dynamics. However, in the introduction, relating statements are rare, and the authors paid more attention to some unrelated points. For example, the soil types previous REOs studies has been used. Is the soil type (Andisols) very important? I do not think so. In my opinion, the authors should show us the shortage in REOs labeling studies (i.e., overlooks the effects of labeling process and sieving method) and its importance in assessing aggregate turnover dynamic and soil C dynamics, possible effects of labeling process and sieving method on aggregate turnover and soil C dynamics and how do them, and the potential relationships between aggregate turnover and soil C dynamics.

L39-42 why do you mention the concept of "humus"? does your study involve the chemical stability of SOM? the POM and HF you studied are fractioned by physical and density fraction method, not by acid or alkaline or thermal hydrolysis, right? Although they have different functions, they are commonly regarded from a perspective of physical stability/protection.

**Reply1:** We gratefully appreciate for your valuable comment. In the Introduction section, We have summarized comment1 as follows:

(1) The introduction does not fit the topic;

(2) Whether the soil type (Andisols) is important for the manuscript's results;

(3) Since 'humus' is not mentioned in the manuscript, the introduction to 'humus' should not be included in the introduction.

we have

(1) restructured introduction section from labeling and sieving processes on aggregate turnover and organic carbon dynamics, to make the introduction section more relevant to the research topic;

(2) added the explanation about the importance of soil types.

Soil type is considered to influence the labelling effect of aggregates, the difference in REO content between soil aggregate fractions was more evident in coarse textured soils (Sandstone, Granite and Alluvium) than in fine textured soils (Red clay, Basalt and Loess). However, REE oxides have been used in limited soil types, such as Alfisol in the studies of De Gryze et al. (2006) and Morris et al. (2019); Ultisols, Mollisols, and Inceptisols in the studies of Liu et al. (2019); and Vertisols in the study of Rahman et al. (2019).

(3) deleted unnecessary description of the relationship between aggregate turnover and soil organic carbon dynamics;

(**Introduction section** is shown in revised manuscript **L25-L88**).

**L25-L88:** Soil structure is a crucial ecosystem service essential for maintaining physical, chemical, and biological processes in the soil. Soil aggregates, clusters of soil particles that adhere to soil organic components, are the fundamental units of soil structure. Soil aggregate dynamics involve aggregate formation, stabilization, and breakdown processes (Oades, 1991; Six et al., 2004). However, evidence shows that soil structure is never in a stable state but is constantly changing (Leij et al., 2002). Studies have suggested that this dynamic behavior mediates, to a large degree, soil organic matter (SOM) turnover and storage (Six et al., 1998; Plante and McGill, 2002). Although considerable studies have investigated soil aggregate dynamics, the life cycle of an aggregate and its impact on microbial-mediated C cycling remain elusive. Hence, quantifying soil aggregate turnover and SOM dynamics is essential to advance the understanding of SOM dynamics (Plante et al., 2002; De Gryze et al., 2006; Peng et al., 2017) and provide a better prediction of the response of the soil system to management practices.

Over the past 40 years, few studies have monitored soil aggregate dynamics using tracers. Indeed, stable isotopes (e.g., 137Cs, 210Pb, 7Be, and 234Th) were first used as tracers to monitor soil aggregate dynamics (Brown, 1981; Ritchie, 1990; Wallbrink, 1993). Then, Olmez (1994) proposed a soil particle labeling method for the diffusion of Au and Ag into sediment crystal lattice by high-temperature heating. However, heating may greatly change the chemical composition of the soil, especially soil clay and soil organic matter, making this method disadvantageous. Others like Staricka et al. (1992) mixed 1-3 mm ceramic spheres into the soil as tracers and found that they incorporated much more rapidly into macroaggregates (>40 mm) than into microaggregates (12-40 mm). However, All these methods have different drawbacks in labeling soil aggregates. Therefore, rare earth element oxides (REE oxides) were proposed to monitor the dynamics of soil aggregates, as they are harmless to the environment and characterized by small particle size (<5 μm), similar physicochemical properties, low background concentrations, limited mobility, and low solubility, resulting in high sensitivity analysis, convenient monitoring results, and low-cost measurement and providing excellent properties as a potential mineral tracer (Tyler, 2004; De Gryze et al., 2006; Peng et al., 2017).

As mentioned above, the REE oxides strongly bind with silty and, in particular, clay soil particles are incorporated into soil aggregates (Zhang et al., 2001), explaining the widespread use of REE oxides as tracers of soil aggregates. Indeed, To obtain reliable results, soil aggregate tracers must be as unaffected and exhibit uniform states in soil. De Gryze et al. (2006) used the mean weight diameter (MWD) to determine the effect of REE oxides addition and labeling processes on soil aggregate,showed no evidence that the tracer affected microbial activity or initial aggregate size distribution. At the same time, calculated the rare-earth element (REE) enrichment percentages in each aggregate size,found the coefficients of variation of the rare-earth element

measurements in eight spatially separate samples were acceptably small and the recovery rates large. From this, Liu et al. (2019) showed no evidence of potential effects of REE oxide types and addition on aggregate recovery rate and aggregate distribution in different soil types. Meanwhile, Morris et al.(2019) provided qualitative snapshots of eight REE oxides labeled aggregate thin sections through the X-ray fluorescence data, which also consistent with the results reported in the previous studies. The above studies demonstrate that REE oxides addition and labeling processes have little effect on soil aggregates, but few studies have concerned the effect of REE oxides addition and labeling processes on soil organic carbon dynamics.

Soil total organic carbon (TOC), occluded organic carbon(oPOC) and heavy fraction(HF) are regarded as slow fractions of soil organic carbon because they change slowly over time because of their large contents, protected by aggregates and associated with minerals (Franzluebbers et al., 1999;Riggs et al.,2015;Marín-Spiotta et al., 2008). In contrast, dissolved organic carbon (DOC) and microbial biomass carbon are considered as active organic carbon fractions that change seasonally (Wilson and Xenopoulos, 2008; Babur and Dindaroglu, 2020) and influence aggregate turnover(Murugan et al., 2019; Bucka et al., 2019). Similarly, free particulate organic carbon (POC) are considered as intermediate organic carbon fractions for changes of soil organic carbon with time that provide substrates for microbes (Witzgall et al., 2021) and also influence aggregation (Bucka et al., 2019). Because the aggregate distribution and aggregate turnover influence soil organic carbon (Six et al., 1998), determination of soil organic fractions provides important information on soil organic carbon sequestration and mineralization with aggregate turnover(Qiu et al., 2023).

The soil aggregate and associated carbon (C) fractions, however, are also influenced by the sieving method (Kemper and Rosenau, 1986; Whalen and Chang, 2002). The most common method of separating aggregates from bulk soil is the wet sieving method, and the wet sieving method is widely used to determine size distribution and stability of aggregates caused by raindrop impact on dry soil causing slaking and the gas pressure inside aggregates (Elliott, 1986; Cambardella and Elliott, 1993). Compared with wet sieving method, The dry sieving method is based on the mechanical impact on soil structure during shaking the samples in a sieve tower, either by hand (Bach & Hofmockel, 2014) or using a mechanical sieve shaker(Nahidan & Nourbakhsh, 2018). As a result, dry sieving method for determining microbial biomass and activities that include water-soluble C and N in aggregates is receiving increased attention because of less destruction to the physical habitat of microbial communities(Kooch et al., 2022). Although information on soil organic carbon fractions is available (Kemper and Rosenau,1986), little is known about variations in soil organic carbon fractions with aggregate size fractions separated by dry and wet sieving methods.

On the other hand, REE oxides have been used in limited soil types, such as Alfisol in the studies of De Gryze et al. (2006) and Morris et al. (2019); Ultisols, Mollisols, and Inceptisols in the studies of Liu et al. (2019); and Vertisols in the study of Rahman et al. (2019). Andisols, the major soil type in Japan, are characterized by high contents of short-range-order (SRO) minerals and organometal complexes, very low bulk density, and a physically stable aggregate structure (Takahashi et al. 2016). A lack of study still exists on the use of REE oxides as tracers to label Andisols and the effect of the labeling process on Andisols organic carbon dynamics. Therefore, according to Peng et al. (2019), the study was divided into the labeling - sieving stage and the Recombination - incubation stage, for both short- and long-term effects, aims to assess (1)   the effect of the labeling   and sieving processes on soil aggregate turnover, (2) the effect of the labeling   and sieving processes on soil organic carbon fractions , and (3) the impact on the quantitative analysis of the relationship between organic carbon and aggregate dynamics. Unlike previous studies that have focused on the effects of REE oxides labeling and sieving methods on soil aggregate size fractions, this paper provides the perspective of soil organic carbon fraction for the feasibility of using REE oxides as Andisols aggregate tracers.

**Materials and methods**

**Comment 2:** more details of the soil properties show be shown, such as SOC, soil texture and etcs.

**Reply2:** We gratefully appreciate for your valuable comment. We provide more soil properties in 2.1Soil characteristics in revised manuscript ( *2.1 Soil characteristics,* **L98-L101**).

**L98-L101:** Prior to the labeling, sieving and incubation processes, the soil properties were: bulk density:0.78g cm$^{-3}$; soil pH: 6.50; total organic carbon (TOC): 53.21 mg g$^{-1}$, total N: 4.54 mg g$^{-1}$ , short range order mineral(SRO: allophane + ferrihydrite): 168 mg g$^{-1}$; Sand (2.0–0.05 mm): 23.92%; Silt (0.05–0.002 mm): 31.02%; Clay (<0.002 mm): 45.06%.

**Comment 3:** L84-85: "soil sampled by a core at five random locations". What's the diameter of soil core? How large the region of soil sampling? Given that soil is highly heterogenous, how representative are the five cores?

**Reply3:** Thank you for your reminding. The diameter of soil core is 5cm. The tillage site is a relatively smaller experimental field than normal farmland, the size of the experimental field is 22.5m × 14m, and well organized by The National Agriculture and Food Research Organization, Japan. They carefully design and manage their experimental fields to avoid heterogeneity in the tillage layers. We ensured during our sampling that the thickness of the tillage layer was also nearly uniform. Wagai et al.

(2013) and Asano and Wagai (2015) also used only six or five samples from same experimental field. Following your comment, we added details about the experimental field in the revised manuscript.( 2.1 Soil characteristics, L94-L106).

**L94-L106:**The soil used in this study was sampled from a typical and one of the most well-characterized allophanic Andisols from the experimental field of the National Institute for Agro-Environmental Sciences in Tsukuba (TKB), Honshu Island, Japan (36° 01′ N, 140° 07′ E, 21 m a.s.l). The tillage site is a relatively smaller experimental field than normal farmland, the size of the experimental field is 22.5m×14m, and well organized by The National Agriculture and Food Research Organization, Japan. This area has a typical subtropical maritime monsoon climate with annual rainfall and mean temperature of 1,282.9 mm and 13.8°C (during 1981–2010), respectively. The soil is derived from volcanic ash deposits and classified as Silandic Hydric Andosols in WRB (IUSS Working Group WRB, 2006), with the mineralogy being dominated by SRO minerals (allophane/imogolite and ferrihydrite), gibbsite, kaolinite, chlorite, hydroxy-interlayered vermiculite, mica, cristobalite, quartz, and feldspar (Asano and Wagai, 2014). The soil samples were collected from 0-20 cm surface layers by a 5cm diameter core at five random locations to prepare one composite sample, which was air-dried and broken by hand to pass through a 2-mm sieve. Prior to the labeling, sieving and incubation processes, the soil properties were: bulk density:0.78g cm-3; soil pH: 6.50; total organic carbon (TOC): 53.21 mg g−1, total N: 4.54 mg g−1 , short range order mineral(SRO: allophane + ferrihydrite): 168 mg g−1; Sand (2.0–0.05 mm): 23.92%; Silt (0.05–0.002 mm): 31.02%; Clay (<0.002 mm): 45.06%.

**Comment 4:**
(1)  L87: you separated three aggregate size classes (0.25-2, 0.053-0.25, <0.053mm), why there have four REOs? One is redundant?
(2)  L95: I doubt that you can broke down to pass the soil through a 2-mm sieve just by hand without other tools after oven-dried. The wet-dry cycle of labeling process had clumped the soil.
(3)  L114: how do you add the 13 ml of ultrapure water to avoid the rewet effect on aggregate turnover?
(4)  L119-120: more details of the incubation. The top of box is open or close?
**Reply4:** We feel sorry for the inconvenience brought to the reviewer. In the previous manuscript, the REE oxides characteristic, labelling, sieving and incubation processes were all placed in *2.2 REO-labeled aggregates*, resulting in a confusing and unreadable experimental design.

Therefore, based on comments, we have restructured the *2.3 experimental design* section in revised manuscript, to bring it more consistent with the topic, and added more details to improve the reader's understanding.

**(1)L87: you separated three aggregate size classes (0.25-2, 0.053-0.25, <0.053mm), why there have four REOs? One is redundant?**

(1) We feel really sorry for our carelessness. Three REE oxides were selected as aggregate tracers, the 2-0.25 mm fraction labeled by Gd2O3 (A), 0.25-0.053 mm fraction labeled by Sm2O3 (B), and <0.053 mm fraction labeled by Nd2O3 (C). The REE oxides characteristics is described in *2.2 REE oxides Characteristics* (**L105-L109**).

**L105-L109:** Three REE oxides (samarium oxide, Sm2O3; neodymium oxide, Nd2O3; and gadolinium oxide, Gd2O3) were purchased from FUJIFILM Wako Pure Chemical Corporation, Osaka, Japan. The median diameter of the powder (D50) ranged between 1.2 and 3.6 μm. The particle density of the material ranged between 6.5 and 7.6 Mg m−3. The background levels of these rare earth elements in the soil were 22.00, 4.70, and 4.80 mg kg−1 for Nd2O3, Sm2O3, and Gd2O3, respectively.

**(2)L95: I doubt that you can broke down to pass the soil through a 2-mm sieve just by hand without other tools after oven-dried. The wet-dry cycle of labeling process had clumped the soil.**

(2) Thank you for your rigorous consideration. As much as reviewer thinks so, there are many clods of soil that are difficult to break up by hand due to complete drying. Therefore, when the samples were dry and not sticky, the bulks were gently broken by hand with gloves. The details were added in *2.3.1 Labeling process*(**L130-L135**).

**L130-L135:** The REE oxides labeling process has been described elsewhere (Peng et al., 2017). Briefly, each REE oxides was suspended in ultra pure water by vortex mixing at a concentration of 300 mg kg−1. 500g soil was continuously mixed and sprayed slowly with the REE oxides suspensions to homogenize labeling, the labeled soil was then stored at 4 ℃ for 7 days to allow water equilibration with minimal microbial activity, followed by incubation at 25 ℃ for 7 days. Next, The soil was dried at 40°C for 48 hours, during this period, when the samples were dry and not sticky, the bulks were gently broken by hand with gloves,   to pass through a 2-mm sieve.

**(3)L114: how do you add the 13 ml of ultrapure water to avoid the rewet effect on aggregate turnover?**

(3) We feel sorry that we did not provide enough information in previous manuscript.
we considered that the rewet effect on aggregate turnover may not need to be avoided. The transformation paths of three aggregate fractions were divided into (1) turnover directly caused by the labeling and sieving processes (at 0 days incubation); and (2) turnover caused by soil microorganisms during the incubation process (at 7,14,21,28 days incubation). The effect of the labeling, sieving and recombination processes on aggregate turnover was represented, by the turnover pathways of the aggregates at 0d

in incubation. This section was described in *2.3 Experimental design*(**L111-L128**):

**L111-L128:** A series of experiments were conducted in this study. First, The feasibility of REE oxides as tracers to track Andisols aggregate turnover was determined. Then, we divide the effects of REE oxides on Andisols aggregate turnover and organic carbon dynamics into two processes: labeling and sieving processes and incubation process. In the labeling and sieving processes, REE oxides addition, labeling method and sieving method are the main causes of soil organic carbon and aggregate turnover. And in the incubation process variations in soil organic carbon dynamics and aggregate turnover caused by initial soil organic carbon fractions differences and the soil microbial.

Five treatments were designed as follows: (1) soil without REE oxides labeling and sieving processes (background treatment, BG), (2) soil with dry sieving and REE oxides labeling (REO-labeled and dry sieved treatment, REO-D), (3) soil with wet sieving and REE oxides labeling (REO-labeled and wet sieved treatment, REO-W), (4) soil with dry sieving but without REE oxides labeling (dry sieved treatment, CK-D), and (5) soil with wet sieving but without REE oxides labeling (wet sieved treatment, CK-W). Where (1) soil without REE oxides labeling and sieving processes (background treatment, BG) is a soil sample that passed a 2 mm sieve, served as a control without any processes. and the other treatments were subjected to labeling, sieving and recombination processes.In (4) and (5), the soil samples were subjected to the labeling process without the addition of REE oxides, followed by dry and wet sieving to obtain dry-sieving aggregates and wet-sieving aggregates,and finally recombined as CK-D treatment and CK-W treatment,which served as a control without REE oxides addition. For (2) and (3), the REE oxides labeled soil samples were sieved dry and wet to obtain REE oxides labeled dry sieving aggregates and REE oxides labeled dry sieving aggregates, which were finally recombined as REO-D treatment and REO-W treatment. Each treatment replicated three times.

**(4)L119-120: more details of the incubation. The top of box is open or close?**

(4) We gratefully appreciate for your valuable comment. Following your suggestion, more details about the incubation process were added in *2.3.4 Incubation process*(**L159-L166**).

**L159-L166:** Five different treatments samples were gently packed into a PVC box (20 cm×20 cm×5 cm) with a flat platen to achieve a bulk density of 0.78g cm−3. Afterwards, the boxes were sealed with cling film, and evenly perforated with pins, to ensure gas exchange and to minimize water loss. The boxes were incubated at 25℃ for 28 days in incubator(EYELA LTI-601SD),and the moisture content was maintained at 60% water-holding capacity    for the whole incubation process by weighing every two days and supplying the lost water using a dropper. The aggregate turnover and SOC fractions were measured by destructively harvesting batches of soil

on 0, 7, 14, 21, and 28 days of incubation. On the one hand, the feasibility of using REE oxides as tracers for aggregate turnover process was verified, and, on the other hand, The short- and long-term effects of the REE oxides labeling and sieving process on soil organic carbon dynamics were analyzed. All treatments were replicated three times.

**Comment 5:**L175: where is the calculation of aggregate turnover rate? I can not find in the Appendix S1.

**Reply5:** We feel sorry for the inconvenience brought to the reviewer. In revised manuscript, we have (1) introduced the calculation procedure in *2.5.1 Calculation of soil aggregate turnover*; (2) inserted the schematic diagram of aggregate turnover, to make it easier for readers to understand(**L209-L250**).

**L209-L250:** The mathematical description of aggregate turnover is considered a multidimensional extension of a simple first-order linear compartment model (De Gryze et al., 2006). Briefly, the current study tracked three different aggregate fractions, viz., A (0.25- to 2-mm fraction), B (0.053- to 0.25-mm fraction), and C (<0.053-mm fraction). The three aggregate fractions can produce three breakdown pathways from larger aggregate fractions into smaller aggregate fractions (a–c) and three buildup pathways from smaller into larger aggregate fractions (d–f). The mass transfer and turnover rate for each aggregate fraction were modified following the method in Peng et al. (2017).

[Figure]

**Figure 2: The 6 possible transformation pathways of aggregate buildup (a－c) and breakdown (d－e) process among the three aggregates fraction. A, B, and C represent 0.25－2 mm, 0.053－0.25 mm and < 0.053 mm aggregates, respectively.**

The mass of soil transfers along each pathway from time t1 to t2 can be described with a discrete transfer matrix K(t2-t1)(Peng et al., 2017):

$$K_{(t_2-t_1)} = \begin{bmatrix} 1-a-b & d & f \\ a & 1-c-d & e \\ b & c & 1-e-f \end{bmatrix} \qquad (1)$$

where a to l are the changes of proportions of REE oxides relating the specific pathways in Figure 2 from time t1 to t2, which is equivalent to the changes of proportions of aggregates relating the specific pathways.

First, we gain the REE oxide concentrations of different aggregate fractions at time t1 as follows:

$$REO_{con.(t)} = \begin{bmatrix} [Gd_A] & [Sm_A] & [Nd_A] \\ [Gd_B] & [Sm_B] & [Nd_B] \\ [Gd_C] & [Sm_C] & [Nd_C] \end{bmatrix} \qquad (2)$$

where, e.g., $Gd_A$ is the concentration of Gd in small macroaggregate (A,0.25-2.00 mm) fraction. The amount of aggregates at time steps t can be described by vectors S(t) :

$$S(t) = \begin{bmatrix} A(t) \\ B(t) \\ C(t) \end{bmatrix} \qquad (3)$$

where the A, B and D represent the amounts of small macroaggregates (0.25-2 mm), microaggregates (0.053-0.25 mm), and silt and clay sized aggregates(<0.053 mm), respectively.

The absolute REE oxide amounts in the three aggregate fractions are:

$$REO_{amo.(t)} = \begin{bmatrix} A(t)[Gd_A] & A(t)[Sm_A] & A(t)[Nd_A] \\ B(t)[Gd_B] & B(t)[Sm_B] & B(t)[Nd_B] \\ C(t)[Gd_C] & C(t)[Sm_C] & C(t)[Nd_C] \end{bmatrix} \qquad (4)$$

When the absolute tracer amounts in aggregates is assumed during transfer between time steps t1 and t2, their relationship can then be described as follows:

$$REO_{amo.(t2)} = K(t_2-t_1)REO_{amo.(t1)} \qquad (5)$$

Consequently, the transformation matrix K(t2-t1) can be calculated:

$$K(t_2-t_1) = REO_{amo.(t1)}^{-1} REO_{amo.(t2)} \qquad (6)$$

where the K(t2-t1) indicates the change in the proportions of aggregates falling into sizes A, B or C between time steps t1 and t2 and f indicates the pathway of C fraction buildup into A fraction, while b indicates the pathway of A fraction breakdown into C fraction (Figure 2).

Because the Andisols samples were subjected to the labeling process, the sieving

process, and the recombination process, and finally to incubation, The labeling, sieving and recombination processes have a destructive effect on aggregates. Soil samples obtained from 7-, 14-, 21- and 28-days incubation included both the labeling-sieving and recombination processes and the incubation process($K_{tx}$), whereas samples from 0 day incubation included only the labeling, sieving and recombination processes ($K_{t0}$), then the contribution of the incubation effect to aggregate turnover is calculated as:

$$K_{inc} = K_{tx} - K_{t0} \qquad (7)$$

Finally, the predicted aggregates fractions were calculated as:

$$A_{t2} = (1 - a - b)A(t_1) + dB(t_1) + fC(t_1) \qquad (8)$$

$$B_{t2} = aA(t_1) + (1 - c - d)B(t_1) + eC(t_1) \qquad (9)$$

$$C_{t2} = bA(t_1) + cB(t_1) + (1 - e - f)C(t_1) \qquad (10)$$

**Results**

**Comment 6:** The authors showed lots of information in the text, without emphasizing the important information associated with the topic. For example, the effectivity of REOs labeling in tracing aggregate turnover has been widely proved, it is not an important information here. So, Fig.3 can be put in appendix rather than in text, and associated text should be more concise. Figs. 1 and 2 also can be merged.

**Reply6:** We gratefully appreciate for your valuable comment. Following your suggestion, we restructured the result section to incorporate 1) the effects of labelling, sieving and recombination processes (0d incubation) on organic matter; 2) the effects of labeling, sieving and recombination processes (0d incubation) on the distribution of agglomerates; 3) to verify the feasibility of REE oxides as tracers for Andisols; 4) the turnover pathways of aggregates during incubation; 5) the organic matter dynamics during incubation. This is actually the process of our experiment, not the validation of the research topic.

In the revised manuscript, to fit the research topic, we restructured the RESULT section as 1) The effect of labeling and sieving processes on Andisols aggregate turnover; 2) The effect of labeling and sieving processes on soil organic carbon fractions; 3) The effects on the quantitative study of the relationship between aggregate turnover and organic carbon dynamics. The modified Result section is as follows:

In response to your question about the importance of the verification of the feasibility of using REE oxides as aggregate tracers, we hope to be able to provide an explanation. In previous studies we have found that REE oxides do not track 2-5mm fractions of Andisols very well during labeling (relevant manuscripts have been submitted and we hope that reviewers could continue to be interested in our studies). Therefore, in this study we needed to first ensure that REE oxides could be used as tracers to track Andisols <2mm fractions aggregate turnover, and then, use REE oxides to quantify aggregate turnover.

**Comment 7:** L194:why the format of Table 1 differs from other tables? And I feel uncomfortable with the unit in Table 1. g kg-1soil for TOC, fPOM, oPOM and HF and mg kg-1soil for MBC and DOC are more commonly used.

**Reply7:** Thank you so much for your careful check. As less attention has been paid to the effects of REE oxides addition, labelling, sieving and recombination processes on soil organic carbon in previous studies. Therefore, in this study, five treatments were designed as follows: (1) soil without REE oxides labeling and sieving processes (background treatment, BG), (2) soil with dry sieving and REE oxides labeling (REO-labeled and dry sieved treatment, REO-D), (3) soil with wet sieving and REE oxides labeling (REO-labeled and wet sieved treatment, REO-W), (4) soil with dry sieving but without REE oxides labeling (dry sieved treatment, CK-D), and (5) soil with wet sieving but without REE oxides labeling (wet sieved treatment, CK-W). Based on this design, we were able to determine the effect of REE oxides addition on the organic carbon fraction of soil aggregates by comparing the REO-D treatment with the CK-D treatment, the REO-W treatment with the CK-W treatment, and the effect of labeling and sieving processes on the organic matter fraction of the soil by comparing the BG treatment with other treatments.

Following your suggestion, g kg-1soil for TOC, fPOM, oPOM and HF and mg kg-1soil for MBC and DOC were used in Table3, and the relevant descriptions of the table were revised accordingly (3.2.1 The effect of labeling and sieving processes on soil organic carbon fractions after labeling and sieving processes, **L361-L384**).

**L361-L384:** The effects of the REE oxides addition, labeling process and sieving process on soil organic carbon were evaluated by assessing the differences in the organic carbon fractions of different treatments at the 0-day incubation. The TOC, oPOC and HF fractions were not significantly different between treatments (P>0.05).

The TOC was maintained at 4.84 -5.01 g/kg soil, oPOC fraction content at 2.54-2.67 g/kg soil and HF at    35.44-37.07 g/kg soil.

Table 3. The mass content of organic carbon fractions in different treatments

| Samples | TOC/g kg$^{-1}$ | MBC/mg kg$^{-1}$ | DOC/mg kg$^{-1}$ | fPOC/g kg$^{-1}$ | oPOC/g kg$^{-1}$ | HF/g kg$^{-1}$ |
|---|---|---|---|---|---|---|
| BG | 48.88 (0.64)$^a$ | 62.00 (1.50)$^a$ | 50.93 (1.00)$^a$ | 1.95 (0.30)$^a$ | 2.54 (0.38)$^a$ | 36.16 (0.21)$^a$ |
| CK-D | 49.56 (0.27)$^a$ | 52.25 (1.60)$^b$ | 30.05 (4.09)$^b$ | 1.03 (0.25)$^b$ | 2.67 (0.51)$^a$ | 36.71 (0.22)$^a$ |
| CK-W | 49.59 (0.032)$^a$ | 38.61 (1.95 )$^c$ | 44.75 (1.13)$^a$ | 0.37 (0.02)$^c$ | 2.62 (0.09)$^a$ | 35.44 (0.15)$^a$ |
| REO-D | 48.44 (0.01)$^a$ | 53.75 (1.90)$^b$ | 37.52 (1.83)$^b$ | 0.99 (0.04)$^b$ | 2.62 (0.17)$^a$ | 36.26 (0.14)$^a$ |
| REO-W | 50.11 (0.43)$^a$ | 39.82 (0.37)$^c$ | 41.35 (0.14)$^a$ | 0.35 (0.03)$^c$ | 2.57 (0.25)$^a$ | 37.07 (0.18)$^a$ |

Different lowercase letters denote significant differences at $P < 0.05$ between different treatments under the same soil organic carbon fractions ($P < 0.05$).

The effect of REE oxides addition on the soil organic carbon fraction was compared by treatments under the same sieving method, such as the dry sieving (REO-D vs. CK-D) treatments and the wet sieving(REO-W vs. CK-W) treatments. The addition of REE oxides exerted no significant effect on SOM fractions. The MBC, DOC, and fPOC showed no significant differences between REO-W and CK-W treatments (P > 0.05); the same result was also observed in the dry sieving aggregate recombination soil, where the MBC, DOC, and fPOC showed no significant differences between REO-D and CK-D treatments (P > 0.05).

The effects of sieving processes on soil organic carbon fractions were estimated by comparing the dry sieving (REO-D and CK-D) treatments with the wet sieving (REO-W and CK-W) treatments. The MBC content was significantly higher in the dry sieving (REO-D and CK-D) treatments compared to the wet sieving (REO-W and CK-W) treatments by 34.98%-35.33% (p<0.05), while the fPOC content was also significantly higher by 176.37%-178. 31% (p<0.05). While DOC content was 10.20%-48.92% higher in the wet sieving treatments (REO-W and CK-W) compared to the dry sieving (REO-D and CK-D) treatments.

The effects of the labeling and sieving process on soil organic carbon required to compare the BG treatment with the dry sieving (REO-D and CK-D) treatment and the wet sieving (REO-W and CK-W) treatment, respectively. The dry sieving (REO-D and CK-D) treatment significantly reduced the DOC and MBC content by ~26.32%‒41.00% and 13.31%‒15.73%, respectively (P < 0.05), and also the fPOC content by 46.81%‒49.22% (P < 0.05). The wet sieving (REO-W and CK-W) treatment exerted stronger effects than dry sieving treatments on MBC and fPOC, with MBC decreasing by 35.77%‒37.72% and fPOC decreasing by 80.89%‒81.62%, whereas the DOC content in the wet sieving treatment was not significantly different from that in BG (P > 0.05).

**Comment 8:** Fig.7: Why do not show the absolute value of soil C fractions such as MBC and DOC? I think use the absolute value is more clearly than the proportions of them in SOC (the values are too low) to assess the effects of REOs labeling and sieving methods.

**Reply 8:** We feel sorry for the inconvenience brought to the reviewer. Following your suggestion, we have replaced the proportions of OC fractions in SOC to the absolute value in figure and revised description of the figures.

As REE oxides addition had no significant effect on the initial state of the soil organic carbon fraction, the REO-W and CK-W treatments were combined into the wet sieving treatment, and the REO-D and CK-D treatments were combined into the dry sieving treatment when comparing long-term effects. The proportions of SOC fractions at different incubation days are shown in Figure 8.

[Figure]

**a** MBC fractions          **b** DOC fraction

[Figure]

**c** fPOC&oPOC    fractions

**d** TOC&HF fraction

**Figure 8: The proportion of different carbon fractions in recombined soils with incubation time.**

The differences in soil organic carbon during incubation caused by labeling and sieving processes were developed by comparing the treatments with the BG treatment. Where, MBC, the small fractions of SOC, remained stable in dry sieving recombined soil at 51.91mg kg-1-54.54mg kg-1 compared with BG treatment during incubation. However, in wet sieving aggregate recombined soil, MBC fraction showed different trends, with slowly decreasing from 39.82mg kg-1 to 34.33mg kg-1 during incubation process. The DOC fraction in both dry sieving and wet sieving recombined soil showed oscillating trends and remained in the range of 30.05mg kg$^{-1}$-46.71mg kg$^{-1}$ from 0 to 28 days.

The oPOC fraction, which is believed to be protected by aggregates, exhibited an decreasing trend throughout the incubation period. The oPOC fractionThe oPOC in dry aggregate recombined soil first decreased from 2.62g kg$^{-1}$ to 2.24g kg$^{-1}$ in 0-7days, and then slowly increasing to 2.46g kg$^{-1}$ during 28 days incubation, as well as oPOC in wet aggregate recombined soil, decreasing from 2.57g kg-1 to 1.9g kg-1. Without labeling, sieving and recombination processes, The oPOC fractions remained stable during incubation, within 3.54g kg$^{-1}$ -3.42g kg$^{-1}$.

In contrast to the oPOC fraction, the fPOC fraction in dry sieving recombined soil increased from 1.03g kg$^{-1}$ to 1.35g kg$^{-1}$ through the first 7 days and then slowly decreasing to 0.98g kg-1 from 7 to 28 days. As same as the wet sieving recombined soil, the fPOC fraction showed a low content in 0 days as the fPOC was removed during the wet sieving process, and the fPOC fraction sharply increasing from 0.36 g kg-1 to 1.18 g kg-1 in 0-7days, followed by a slow increasing to 1.77 g kg$^{-1}$ during 7－28 days. While the BG treatment showed decreasing from 1.95 g kg$^{-1}$ to 1.82 g kg-1 in 0-7 days, and then gradually decreased to 1.70 g kg$^{-1}$ after 28 days.

The HF fractions showed remarkable similarity in both recombined soils, declining

incubation process, even though HF did not show a clear trend in the BG treatment. The HF fraction decreased more significantly in wet sieving recombined soil, especially in 0-7days, from 36.07 g kg$^{-1}$ to 30.8 g kg$^{-1}$, and then remained relatively stable. TOC showed a decreasing trend during incubation. TOC showed a decreasing trend during incubation, with BG treatment and dry sieving recombined soil showing a similar decreasing trend, and the wet sieving recombined soil showed a greater decrease from 50.11 g kg$^{-1}$ to 46.36 g kg$^{-1}$ on 0-7 days and then remained slowly decreasing.

**Discussion**

**Comment 9:** I completely agree with the comments of another review about the discussion.

Besides, for section 4.3, authors paid more attention to discuss the aggregate turnover dynamics, where is the discussion of comparing the effects of labeling process and sieving method on aggregate turnover? I think it is the key point needing to be discussed.

**Reply9:** We feel sorry for the inconvenience brought to the reviewer. We have tried too much to illustrate the feasibility of using soil organic carbon fractions and REE oxides to quantify soil organic carbon dynamics and soil aggregate turnover, respectively, and to analyze their relationship. This resulted in the Discussion section being inconsistent with the research topic.

(1) We discussed the effect of labeling and sieving processes on the aggregate turnover from 1)the feasibility of REE oxides as Andisols aggregate tracers; 2) The transformation of aggregate turnover pathways before incubation (0d); 3) The transformation of aggregate turnover pathways during incubation;4) the aggregate turnover rate during incubation(*4.1 Effects of labeling and sieving processes on Andisols aggregate,* L436-L488).

(2) We discussed the effects of the labeling and sieving process on soil organic carbon in terms of 1) the effect of wet sieving on soil organic carbon dynamics; 2) the effect of dry sieving on soil organic carbon dynamics(4.2 Effects of REE oxide labeling and sieving processes on soil organic carbon, L490-L530);

3) We discussed the feasibility of using soil organic carbon fractions to analyze the relationship between organic carbon dynamics and aggregate turnover (Effects of the quantitative study of the relationship between aggregate turnover and organic carbon dynamics, L554-L586).

**Comment 10:** L334-338: explaining why wet sieving removes most fPOM. Does the dry sieving reduce fPOM because of the removal of fine roots and debris? The fine roots and debris should be removed before experiment, them were not pick out clean?

**Reply 10:** Thanks for your careful checks,we feel sorry that we did not provide enough information about dry sieving and wet sieving methods. The soil samples were root-picked prior to sieving, so both the wet and dry sieving treatments included fPOC loss from the root-picking process. However, during the wet sieving process, we found some free particulate organic matter floating on the water after sieving and this part of the POC was lost during the sieving process. Therefore, the fPOC of the wet sieving aggregates was significantly lower than that of the dry sieving aggregates and the BG treatment in this study.

**Comment 11:**
(1) Some problems of format at L345, 351, 361.
(2) L394-395: If the relationship between oPOM and aggregate fractions is important, please discuss it more in-depth, not just depict the result. If it is not important in this study, it is not necessary to show.
(3) L415: "The HF generated by aggregate breakdown cannot accumulate into aggregates, allowing it to accumulate on the outside and consequently increase the proportion of fPOM", please attach the reference. I do not think the accumulation of HF can increase fPOM. They are varying in size, density and properties (e.g. C/N), two different concepts. The soil continuum model (Lehmann, J., Kleber, M., 2015. The contentious nature of soil organic matter. Nature 528: 60-68) suggested that fPOM can be degraded into HF with microbial processing, and it is an irreversible process.

**Reply 11:** It is really a giant mistake to the whole quality of our article. We feel sorry for our carelessness. We have corrected it and we also feel great thanks for your point out.Following the suggestion by you and the other anonymous reviewer, we removed the extensive discussion of the relationship between oPOM and aggregate fractions, and restructured the discussion in terms of labelling, sieving processes for 1) aggregate turnover; 2) organic matter dynamics; and 3) quantitative analysis of the relationship between organic matter and aggregates.

In response to L415: "The HF generated by aggregate breakdown cannot accumulate into aggregates, allowing it to accumulate on the outside and consequently increase the proportion of fPOM", not from other literature, but for the quantitative analysis of soil organic matter dynamics in relation to agglomerates during wet sieving, we found an increase in fPOC content and a decrease in oPOC and HF during incubation, and therefore feel that the increase in fPOC may be related to HF. The inaccurate description has been removed following your suggestion. The revised Discussion section is as follows

**4.1 Effects of labeling and sieving processes on Andisols aggregates**

Labeling and sieving processes have no effect on the feasibility of REE oxides as tracers to track aggregate turnover. The relationship between the measured and

predicted results of soil dry sieving aggregates (Figure 3) was consistent with the finding reported by Morris et al.(2019). In the study of Morris et al.(2019), X-ray fluorescence data were used to provide insight into mechanisms operating during aggregate turnover, showing that REE oxides are effective tracers for studying the dry sieving aggregate formation and decomposition dynamics. Also, the close 1:1 line between measured and predicted aggregates (Figure 4) supported Peng et al.'s (2017) findings that REE oxides are effective tracers of wet sieving aggregate dynamics. Whilst REO was an effective tracer for both dry sieving and wet sieving aggregate, there were limitations. The weaker relationship for 0.053-0.25 mm(slope = 0.89) wet sieving aggregates (Figure 4) may be an experimental artefact due to the additional loss or gain of these fraction during the wet sieving procedure.

To elucidate the influence of the labeling and sieving processes on the aggregates turnover, The transformation paths of three aggregate fractions was divided into: (1) turnover directly caused by the labeling and sieving processes (at 0 days incubation); and (2) turnover caused by soil microorganisms during the incubation process (at 7,14,21,28 days incubation). Labeling and sieving processes affect the aggregate turnover pathways before incubation(0 day). In particular, the turnover pathway of wet sieving aggregates (Figure 8 and 10) is consistent with the study in Peng et al. (2017), where at 0 days incubation, soil aggregates were mainly affected by labeling, sieving and recombination processes, producing a large number of aggregate breakdown-formation transformation pathways. In the study by Peng et al. (2017), the 0.25-2mm wet sieving aggregate fraction was used as an example. 26% broken down to 0.053-0.25 aggregate fraction and another 26% broken down to 0.053% aggregate fraction. The effect of wet sieving on soil aggregates has been shown to depend on size, structure, shrinkage and expansion behavior, wettability of soil components, porosity and the spatially heterogeneous distribution of these properties within the aggregates (Baumgartl & Horn, 1993; Chenu et al., 2000; Kaiser et al., 2015). De Gryze et al. (2006) used artificial wet sieve agglomerates and after the sieving and recombination process, 30% of the 0.25-2mm aggregate fraction broken down to 0.053-0.25 aggregate fraction, and 31% broken down to <0.053mm aggregate fraction, this fraction higher than the transformation of turnover pathways in our study;

Another reason might be the different fractions of aggregates. In previous studies, 2-5mm fraction, 0.25-2mm fraction, 0.053-0.25mm fraction and 0.053mm fraction were studied as simplifications of soil structure (Liu et al.,2019), and since Andisols has a small proportion of 2-5mm fractions (Asano et al.,2014). Therefore, in this study, three aggregate fractions (0.25-2mm fraction, 0.053-0.25mm fraction and 0.053mm fraction)were selected. The effect of labeling and sieving on dry sieving aggregates has not been studied before. Morris demonstrated the effect of AMF on aggregates turnover by dry sieving aggregates turnover, but the study only showed the aggregates turnover path after 5 week incubation and no further study of the effect of the labeling

and sieving process was performed. The comparison of sieving methods demonstrated that although wet-sieving aggregate fractions were found to contain more free primary particles and fine-sized aggregates, dry-sieving aggregate fractions contained more aggregates with inclusions of primary particles.

Similarly, the labeling and sieving processes also impact on aggregate turnover pathways during the incubation process. Since the turnover path is the sum of the labeling and sieving effects and the incubation process, the turnover pathways of the incubation process is calculated as the difference between the turnover pathways of different incubation days and the turnover pathways of 0 days incubation (as equation 7). Due to the different topics of the studies, Peng et al. (2017), De Gryze et al. (2006) and M. Halder et al (2022) did not distinguish the wet sieve aggregate turnover pathways. With the same approach to divide the aggregate turnover pathways in Peng et al (2017), With the same approach to divide the aggregate turnover pathways in Peng et al (2017), it was found that during the incubation period, the aggregate turnover mainly occurred in 0-7days, with 16% of the 0.25-2 mm aggregate fraction was broken down to 0.053-0.25 mm aggregate fraction and 13% to 0.053 aggregate fraction. In the following 7-14 days, the proportion of 0.25-2mm aggregate fraction breaking down to 0.053-0.25mm aggregate fraction was -2%, and the proportion breaking down to 0.053 aggregate fraction increased only 1%. Such trends are consistent with our study, and also consistent with the M. Halder et al's (2022) calculations for wet sieving aggregate turnover. In M.Halder    et al.'s(2022) study,

The transformation pathways of 0.25-2 mm aggregate fraction breaking down to 0.053-0.25 mm aggregate fraction was 3%, which was similar to the breaking ratio of 0.25-2 mm aggregate fraction to 0.053-0.25 mm aggregate fraction in this experiment. For the effect of labeling and sieving processes on dry sieving aggregates turnover processes, Morris et al. (2017) classified dry sieving aggregates as 1mm ,0.212mm,0.053m and <0053mm aggregate fractions, and the experiment only reported aggregates turnover paths after 5 weeks of incubation, making it difficult to make comparisons.

The aggregates turnover rate during the incubation is also affected by the labeling and sieving process. The turnover rate of wet sieving aggregates showed the maximum rate from 0-7 days and then gradually decreasing. This is consistent with peng et al (2017),where the 0.25-2 mm fraction showed turnover rates of 4.14%/day at 7 days, then decreasing to 2%/day at 14 days and further decreasing to 1.04%/day at 28 days during incubation. M. Halder et al. (2022) also showed the same trend that the wet sieving aggregates turnover faster at the beginning of the incubation period. The turnover rate of the 0.25-2 mm fraction wet sieving aggregates was 0.14%/day at 14 days, decreasing to 0.035%/day at 28 days of incubation.

**4.2 Effects of REE oxide labeling    and sieving processes on soil organic carbon fractions**

The addition of REE oxides has no effect on soil organic carbon dynamics, which is supported by Zhang et al. (2001), Peng et al. (2017), De Gryzel, and also REE oxides are considered have no effect on soil microorganisms. The effect of labeling and sieving on soil organic carbon is discussed in terms of both (1) the labeling and sieving processes and (2) the incubation process. During labeling and sieving, the fPOC, MBC and DOC contents of the wet sieving recombined soil were significantly lower than those of the dry sieving recombined soil. The initial low content of the fPOC fraction in wet sieving aggregates(Table3) could be explained from two aspects. On the one hand there are several wet-dry cycles during the labeling process, where dry-wet cycles are believed to consume POC (Denef et al., 2001); and on the other hand there is a significant amount of free organic particles (i.e. particles >53 μm that are not enclosed within the aggregates) loss during the wet sieving process ((Haynes, 2000). Wet sieving leads to a loss of total C or total N, especially for soil fractions <250 μm,and the wet-sieving method decreases the nitrogen microbial biomass in comparison to dry sieving (Sainju, 2006).   A recent study comparing the effect of dry and wet sieving on microbial enzymatic activity showed that wet sieving overestimated the potential microbial enzymatic activity in comparison to dry sieving (Bach and Hofmockel, 2014). The DOC fraction, which in the fractionation procedure applied here represents water-extractable SOC, is generally regarded as an active fraction (Zimmermann et al., 2007), in wet sieved recombined soil less than in dry sieved recombined soil,which in line with Udom et al.(2020) suggested that the DOC content in wet-sieved aggregates significant loss   during wetting and slaking in water.

And during the incubation process, the wet sieving aggregate turnover was found affected by soil microorganisms (Blaud et al., 2017) and the rewetting of dry samples during wet sieving was shown to affect the characterization of microbial communities in soil aggregates (Bach et al., 2018),It is consistent with our research that the proportion of the MBC fraction in soil organic carbon shows a decreasing trend with incubation time. In contrast to the MBC fraction, the fPOC fraction in soil organic carbon gradually increased, with other fractions decreased(as shown in Figure 8). These can be explained by(as reviewed by Kaiser et al., 2015) the formation of new interactions between minerals and OM, the killing of microorganisms and shifts in their community structure and an increase of mineral surface acidity and hydrophobicity (possibly affecting aggregate stability; (Ibrahimi et al., 2019)).The degree to which these characteristics are altered by wetting and drying depends on various factors, such as soil depth (Kaiser et al., 2015), land use (Fierer & Schimel, 2003), texture and OM content (Albalasmeh & Ghezzehei, 2014), sodium and calcium cation configuration (Aquino et al., 2011), pH value (Kang & Xing, 2008) and drying speed (Kemper & Rosenau, 1986). All of these influences alone are difficult to quantify, which is why (repeated and/or unnecessary) wetting and drying should be

avoided to preserve the natural properties of soil aggregates and to prevent the creation of artifacts.

The Dry sieving method avoids the changes in physical, chemical and biological properties that can be observed during drying and wetting(Sainju et al.,2003). This is also demonstrated in this experiment, the labeling and dry sieving processes also affect the soil organic carbon fractions, but the effects appeared smaller. In contrast to the wet sieving treatment.The fPOC and DOC fractions increased from 0-7 days with dry sieving treatment(Figures 8b, 8c), which result partially from the decay of some soil organisms after drying (Lund and Goksoyr., 1980; Bottner, 1985) and partially from the release of previously inaccessible organic compounds (Sorensen, 1974; Van Gestel et al., 1991). The increased availability of organic substrates leads to increased microbial activity and increased growth of fungal mycelium and bacterial biomass (Jager and Bruins, 1974), this would explain the increased MBC content of the dry sieving treatment form 0-7days, and it remains stable at around 0.75%SOC (Figure 8a). For this reason, Bach & Hofmockel (2014) indicated that wet sieving process should be used to assess long-term changes in microbial activity and organic carbon fractions, while dry sieving process without rewetting could captured short-term dynamics of soil organic carbon as well. Thus, the effects of labeling and sieving processes on aggregates and organic carbon need to be considered, before using soil organic carbon fractions and REE oxides to quantify organic carbon dynamics and aggregate turnover respectively.

**4.3 Effects of the quantitative study of the relationship between aggregate turnover and organic carbon dynamics**

Further data are needed to support the application of soil organic carbon fractions to quantify soil organic carbon dynamic, with REE oxides to track the aggregates turnover. In previous studies, Peng et al. (2017) analysed the organic carbon dynamics with by adding 13C-labeled glucose to REE oxides labeled soils and determining the 13C content in different aggregate fractions. Subsequently, M. Halder et al. (2022) used eleven organic materials were characterized in terms of nutrient stoichiometry, biochemical features and carbon (C) functional groups, to determine which characteristics of organic materials control soil aggregate turnover. In this study, the correlation between the aggregates turnover and organic carbon fraction dynamics was analyzed. We found that the assessment of the relationship between the turnover pathways of the aggregates and the organic matter dynamics was not affected by labeling and sieving processes (as shown in Table4 and table5). The turnover Pathways of microaggregate were correlated with oPOC and fPOC (p<0.05), which is consistent with the theory of aggregate hierarchy proposed by Tisdall and Oades (1982), where oPOC could act as a nucleus for microaggregates and microaggregates could provide physical protection for oPOC (Pulleman et al.,2004). Also, the effect of MBC and DOC on the aggregate turnover pathways is mainly concentrated in the

<0.053mm fraction. This is consistent with the MEMS framework proposed by courto et al (), where POC decomposed from aggregate turnover were utilised by soil microorganisms, and transformed into DOC which combined with mineral to form Mineral associated organic matter(MAOM, Kleber et al.,2021), MAOM could act as a nucleus for aggregates and promote the formation of soil microaggregates(M.Halder et al., 2021). This interaction results in organic matter dynamics that changes in accordance with soil aggregate turnover, which makes the assessment of the relationship between aggregate turnover pathways and organic matter dynamics independent of labeling and sieving processes. This also explains the fact that Peng et al. (2017) used total turnover pathways to analyze the relationship between aggregate turnover pathways and soil organic matter during incubation, and still obtained good interpretability and correlation.

Labeling and sieving processes affect the assessment of the relationship between aggregate turnover rates and organic matter dynamics. The labeling and sieving processes caused a rapid change in aggregates (Figure 5). Aggregate decomposition during wet sieving is mainly (a) slaking due to compression of entrapped air during wetting (Emerson,1967), (b) microcracing due to differential swelling(Truman et al,1990) and(c) physico-chemical dispersion due to osmotic(Loch,1994), the decomposition of aggregates in the dry sieving process is mainly due to mechanical breakdown (Le Bissonnais, 1996). Aggregate breakdown rates occurring during the sieving process are much higher than aggregate turnover rate during incubation (Peng et al.,2017). In contrast, the turnover rate of aggregates during the incubation process (Figures 6, 7) was smaller. This is consistent with studies by Peng et al. (2017) and M. Halder et al. (2022), where the aggregates turnover was larger before incubation (0days), and then the turnover rate decreased with increasing incubation time. The variation in aggregate turnover rates at different processes leads to different assessments of turnover rates and organic carbon dynamics (labeling, sieving and incubation processes vs. incubation process). At the same time, this study only analyzed the relationship between soil aggregates and organic matter dynamics without the addition of exotic organic carbon. However, for more complex conditions, more research is needed to determine whether REE oxides and soil organic carbon fraction could be used, to quantify the relationship between aggregate turnover and organic carbon dynamics.

**Conclusion**

**Comment 12:** Do not simply repeat the results, but show the main findings and implications.

**Reply 12:** Thank you for your valuable suggestion.

Conclusions based on the research topic and discussion were obtained from aggregate turnover and soil organic carbon dynamics, respectively. The addition of REE oxides

would have no effect on the Andisols aggregate turnover and organic carbon dynamics, but the REE oxides labeling and sieving processes would have effects on soil aggregates and soil organic carbon(**L532-L548**).

**L532-L548:** It was shown that the addition of REE oxides would have no effect on the Andisols aggregate turnover and organic carbon dynamics, but that the REE oxides labeling and sieving process would have effects on soil aggregates and soil organic carbon fractions. The experiments were divided into labeling & sieving processes and incubation process according to the mechanism of aggregate turnover. During the labeling and sieving processes, dry sieving aggregates showed a lower transformation of aggregate turnover pathways before incubation (0 d) compared to wet sieving aggregates. Aggregate turnover mainly occurred in 0-7 days, followed by a gradual decrease of both turnover paths and rates in 7-28 days. However, the labeling and wet sieving process involved several wet-dry cycles that may remove the bacterial cells, resulting in a lower turnover path and rate of wet sieving microaggregates. It is clear that the choice of sieving method was reflected in the different processes of aggregates turnover, and this should be considered in future studies.We suggest to measure the aggregates turnover at 0 days to eliminate the turnover caused by the labeling and sieving processes.

For organic carbon dynamics, the dry sieving treatment retained more MBC and fPOC fractions after the labeling and sieving processes. During the incubation process, the labeling and sieving methods also play a role in the organic carbon dynamics. During the incubation process, the labeling and sieving methods also play a role in the organic carbon    dynamics. The organic carbon dynamics of the dry sieving process are more consistent with those non-labeled & sieving processes soils. Labeling and sieving processes affect the assessment of the relationship between aggregate turnover rates and organic matter dynamics, but not the assessment of the relationship between the transformation of aggregate turnover pathways and organic matter dynamics. As we only studied Andisols organic carbon dynamics and aggregate turnover without the addition of exotic organic carbon, further research is still needed to support the application of soil organic fractions and REE oxides tracers to quantify the relationship between the soil organic carbon dynamics and aggregate turnover.

**Acknowledgments**

We gratefully thanks for the precious time the reviewer spent making constructive remarks.

**L552-L553:** The authors are grateful to the editor and two reviewers for their insightful comments that greatly improve this manuscript.

**Reference**

Based on your suggestions, the Introduction and Discussion sections have been

revised a lot, so we have restructured the Reference section.(**8 Reference**, **L557 - L672**).

We hope you will find our revised manuscript acceptable for publication.
Yours sincerely,
Wang Yike